# Nominality Score Conditioned Time Series Anomaly Detection by Point/Sequential Reconstruction

**Chih-Yu Lai**∗
Department of EECS, MIT
Cambridge, MA 02139
chihyul@mit.edu

**Fan-Keng Sun**
Department of EECS, MIT
Cambridge, MA 02139
fankeng@mit.edu

**Zhengqi Gao**
Department of EECS, MIT
Cambridge, MA 02139
zhengqi@mit.edu

**Jeffrey H. Lang**
Department of EECS, MIT
Cambridge, MA 02139
lang@mit.edu

**Duane S. Boning**
Department of EECS, MIT
Cambridge, MA 02139
boning@mtl.mit.edu

## Abstract

Time series anomaly detection is challenging due to the complexity and variety of patterns that can occur. One major difficulty arises from modeling time-dependent relationships to find contextual anomalies while maintaining detection accuracy for point anomalies. In this paper, we propose a framework for unsupervised time series anomaly detection that utilizes point-based and sequence-based reconstruction models. The point-based model attempts to quantify point anomalies, and the sequence-based model attempts to quantify both point and contextual anomalies. Under the formulation that the observed time point is a two-stage deviated value from a nominal time point, we introduce a *nominality score* calculated from the ratio of a combined value of the reconstruction errors. We derive an *induced anomaly score* by further integrating the nominality score and anomaly score, then theoretically prove the superiority of the induced anomaly score over the original anomaly score under certain conditions. Extensive studies conducted on several public datasets show that the proposed framework outperforms most state-of-the-art baselines for time series anomaly detection.

## 1 Introduction

Time series anomaly detection involves identifying unusual patterns or events in a sequence of data collected over time [1]. This technique is crucial in fields such as finance [2], healthcare [3], manufacturing [4], transportation [5], and more [6]. A comprehensive evaluation of different techniques can be found in [7]. Time series anomaly detection using unsupervised learning approaches is favored by recent studies due to the fact that they don't require labeled data and have better detection of unseen anomalies [8], which is also the approach used in this work. Most unsupervised time series anomaly detection methods involve calculating an anomaly score at each time point and then comparing this score to some threshold. For calculating this score, we can categorize different methods into three main groups: Reconstruction-based methods involve reconstructing the original time series data and comparing the reconstructed data with the actual data [9, 10, 11]. Prediction-based methods involve predicting the next value in the time series and comparing it with the actual value [12, 13, 14]. Dissimilarity-based methods measure the distance between the value obtained from the model and the distribution or cluster of the aggregated data [15, 16, 17, 18, 19, 20]. There can also be hybrid techniques where multiple methods are applied.

---

∗Code available at https://github.com/andrewlai61616/NPSR

37th Conference on Neural Information Processing Systems (NeurIPS 2023).

While the classification approaches for the types of anomalies differ in the literature [6, 7], we will focus on two classes: point anomalies and contextual anomalies. Point anomalies refer to individual data points that significantly deviate from the expected behavior of the time series and can be detected by observing the data at a single time point. Contextual anomalies, on the other hand, refer to data points that deviate from the expected behavior of the time series in a specific context or condition. These anomalies cannot be detected by observing the data at a single time point and can only be detected by observing the contextual information. As a result, point anomalies can be detected using any general anomaly detection technique that does not require temporal information. Such models are called *point-based models* in this work. However, detecting contextual anomalies requires a model that can learn temporal information. Such models require a sequence of input, hence they are termed *sequence-based models* in this work. A time point may contain both point and contextual anomalies. Obviously, contextual anomalies are harder to detect. An important trade-off arises when a model tries to learn time-dependent relationships for detecting contextual anomalies but loses precision or accuracy in finding point anomalies. This trade-off becomes more significant in high-dimensional data, where modeling the temporal relationship is difficult [21].

Our intuition comes from the observation that state-of-the-art methods using sequence-based reconstruction models encounter the point-contextual detection trade-off, resulting in noisy reconstruction results and suboptimal performance. As an alternative, we start by experimenting with point-based reconstruction methods, which exhibit a lower variance as they do not require the modeling of time-dependent relationships. Despite the absence of temporal information, we find that the corresponding anomaly score for a point-based reconstruction model can already yield a competitive performance. To further bridge the gap between point-based and sequence-based models, we introduce a *nominality score* that can be calculated from their outputs and derive an *induced anomaly score* based on the nominality score and the original anomaly score. We find that the induced anomaly score can be superior to the original anomaly score. To provide theoretical proof of our findings, we frame the reconstruction process as a means to fix the anomalies and identify underlying nominal time points and prove that the induced anomaly score can always perform better or just as well as the original anomaly score under certain conditions. We also conduct experiments on several single and multi-entity datasets and demonstrate that our proposed method surpasses the performance of recent state-of-the-art techniques. We coin our method *Nominality score conditioned time series anomaly detection by Point/Sequential Reconstruction* (NPSR).

## 2 Related Works

Reconstruction-based techniques have seen a diversity of approaches over the years. The simplest reconstruction technique involves training a separate model sequentially for each channel using UAE [11]. One type of improvement focuses on network architecture, with notable examples like LSTM-VAE [22], MAD-GAN [23], MSCRED [24], OmniAnomaly [9], and TranAD [25]. Additionally, hybrid architectures like DAGMM [26] and MTAD-GAT [27] have been proposed. Another type aims to improve the anomaly score instead of using the original reconstruction error. Designing this anomaly score involves a considerable amount of art, given the high diversity of methods for calculating it across studies. For instance, USAD uses two weighted reconstruction errors [28], OmniAnomaly employs the "reconstruction probability" as an alternative anomaly score [9], MTAD-GAT combines forecasting error and reconstruction probability [27], and TranAD uses an integrated reconstruction error and discriminator loss as the anomaly score [25]. In the context of network architecture, our method utilizes a straightforward performer-based structure without incorporating any specialized components. Based on our insights into the point-sequential detection tradeoff, our approach stands out by integration of point-based and sequence-based reconstruction errors for competitive performance.

## 3 Methods

### 3.1 Problem Formulation

Let $\mathbf{X} = \{\mathbf{x}_1, ..., \mathbf{x}_T\}$ denote a multivariate time series with $\mathbf{x}_t \in \mathbb{R}^D$, where $T$ is the time length and $D$ is the dimensionality or number of channels. There exists a corresponding set of labels $\mathbf{y} = \{y_1, ..., y_T\}, y_t \in \{0, 1\}$ indicating whether the time point is normal ($y_t = 0$) or anomalous ($y_t = 1$). For a given $\mathbf{X}$, the goal is to yield anomaly scores for all time points $\mathbf{a} = \{a_1, ..., a_T\}, a_t \in \mathbb{R}$ and a corresponding threshold $\theta_a$ such that the predicted labels $\hat{\mathbf{y}} = \{\hat{y}_1, ..., \hat{y}_T\}$, where $\hat{y}_t \triangleq \mathbb{1}_{a_t \geq \theta_a}$, match

**y** as much as possible. To quantify how matched $\hat{\mathbf{y}}$ and $\mathbf{y}$ is, or how good $\mathbf{a}$ is for potentially yielding a good $\hat{\mathbf{y}}$, there are several performance metrics that takes either $\mathbf{a}$ or $\hat{\mathbf{y}}$ into account [11, 12, 29, 30, 31]. This work mainly focuses on the best F1 score (F1*) without point-adjust, also known as the point-wise F1 score, which is defined as the maximum possible F1 score considering all thresholds. (A complete derivation for F1* is covered in Appendix A.)

## 3.2 Nominal Time Series and Two-stage Deviation

We denote the observed data as $\mathbf{X}^0 = \{\mathbf{x}_1^0, ..., \mathbf{x}_T^0\}$, where $\mathbf{x}_t^0 \in \mathbb{R}^D$. Assume that for each $\mathbf{X}^0$, there exists a corresponding underlying *nominal* time series data $\mathbf{X}^* = \{\mathbf{x}_1^*, ..., \mathbf{x}_T^*\}$ that comes from a nominal time-dependent process $\mathbf{x}_t^* = \mathbf{f}^*(t) : \mathbb{N} \to \mathbb{R}^D$. The corresponding total deviation at $t$ ($\Delta \mathbf{x}_t^0$) is defined as $\Delta \mathbf{x}_t^0 \triangleq \mathbf{x}_t^0 - \mathbf{x}_t^*$. We denote $\mathcal{X}^*$ as the set of all possible $\mathbf{x}_t^*$ for all $t \in \{1, ..., T\}$. $\Delta \mathbf{x}_t^0$ can be separated into two additive factors $\Delta \mathbf{x}_t^c$ and $\Delta \mathbf{x}_t^p$, such that, by definition, $\mathbf{x}_t^c = \mathbf{x}_t^* + \Delta \mathbf{x}_t^c$ and $\mathbf{x}_t^0 = \mathbf{x}_t^c + \Delta \mathbf{x}_t^p$. We define $\Delta \mathbf{x}_t^c$ as the *in-distribution deviation*, where $\mathbf{x}_t^c \in \mathcal{X}^*$; and $\Delta \mathbf{x}_t^p$ as the *out-of-distribution deviation*, which is non-zero if and only if $\mathbf{x}_t^0 \notin \mathcal{X}^*$. $\Delta \mathbf{x}_t^p$ can be a means for quantifying the point anomaly, and $\Delta \mathbf{x}_t^c$ can be a means for quantifying the contextual anomaly. This is reasonable, since no matter how large $\Delta \mathbf{x}_t^c$ is, we still have $\mathbf{x}_t^c \in \mathcal{X}^*$, i.e., the in-distribution deviated value $\mathbf{x}_t^c$ is still in the set of all possible nominal time point data, and cannot be detected using a point-based model. On the other hand, it is possible that having learned $\mathcal{X}^*$, a point-based model can negate the deviated value caused by $\Delta \mathbf{x}_t^p$. Fig. 1(a) gives an illustration of the relationships between the variables at time $t$. We clarify this using the example below.

Assume we obtain a dataset from the streaming data of a 2D position sensor, where $\mathbf{x}_t^*, \mathbf{x}_t^c, \mathbf{x}_t^0 \in \mathbb{R}^2$, and we have learned that the nominal time series is the circular movement of a point around the origin with some angular velocity $\omega$ and radius $r$, where $R_{min} \le r \le R_{max}$. Accordingly, we can deduce that $\mathcal{X}^* = \{[x \ y]^T | R_{min}^2 \le x^2 + y^2 \le R_{max}^2\}$ and $\mathbf{x}_t^* = [r \cos \omega t \ \ r \sin \omega t]^T$. One possible cause (among many others) of contextual anomalies might be due to an unexpected change in angular velocity. For instance, failures in the system might lead to a slowdown of the circular movement between $t_1$ and $t_2$, i.e., $\Delta \mathbf{x}_t^c = [r(\cos \omega' t - \cos \omega t) \ \ r(\sin \omega' t - \sin \omega t)]^T$ for $t \in \{t_1, ..., t_2\}$ and $\Delta \mathbf{x}_t^c = \mathbf{0}$ elsewhere. Moreover, noisy measurements of individual time points may induce point anomalies in the observed time series, i.e., $\Delta \mathbf{x}_t^p = [w_{x,t} \ \ w_{y,t}]^T$ where $w_{x,t}$ or $w_{y,t}$ is nonzero such that $\mathbf{x}_t^0 \notin \mathcal{X}^*$ for some $t$. Fig. 1(b)(c) gives an illustration of the above example. The black dots are time points where $\mathbf{x}_t^0 = \mathbf{x}_t^*$ (no anomalies). The blue dots are time points exhibiting a slowdown (contextual anomalies). The red dots are time points with noisy measurements (point anomalies). The purple dots are time points with both slowdown and noisy measurements (point and contextual anomalies). The green dots are time points with noisy measurements but $\mathbf{x}_t^0 \in \mathcal{X}^*$, so are still contextual anomalies since their deviations *cannot* be detected by observing a single time point.

## 3.3 The Nominality Score

Now we conceptualize the *Nominality Score* $N(\cdot)$. Analogous to the anomaly score, $N(\cdot)$ indicates how normal a time point is. A nominality score $N(\cdot)$ is *appropriate* if for every possible $\theta_N > 0$, $\mathbb{P}(N(t) > \theta_N | y_t = 0) > \mathbb{P}(N(t) > \theta_N | y_t = 1)$ for all $t \in \{1, ..., T\}$, i.e., the portion of normal points that has a nominality score larger than $\theta_N$ is strictly larger than the portion of anomaly points

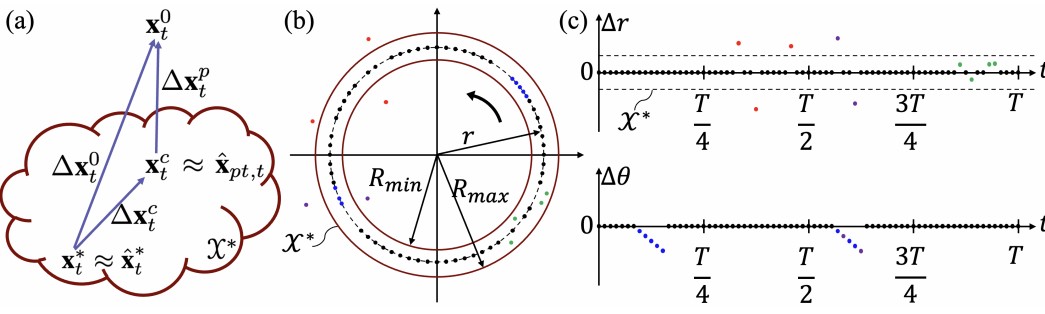

Figure 1: (a) Relationships between variables, (b) observed time series on 2D plane, and (c) radial and angular displacement vs time from nominal time series ($\mathbf{x}_t^*$) for the 2D position sensor example.

that has a nominality score larger than $\theta_N$. There are many ways to define $N(\cdot)$. In this study, we define $N(t)$ as the ratio of the squared L2-norm between $\Delta\mathbf{x}_t^c$ and $\Delta\mathbf{x}_t^0$.

$$N(t) \triangleq \frac{\|\Delta\mathbf{x}_t^c\|_2^2}{\|\Delta\mathbf{x}_t^0\|_2^2} = \frac{\|\Delta\mathbf{x}_t^c\|_2^2}{\|\Delta\mathbf{x}_t^c + \Delta\mathbf{x}_t^p\|_2^2} = \frac{\|\mathbf{x}_t^c - \mathbf{x}_t^*\|_2^2}{\|\mathbf{x}_t^0 - \mathbf{x}_t^*\|_2^2} \tag{1}$$

We provide an example as to when $N(\cdot)$ will be appropriate. In this derivation, we add an $n$ and $a$ in the subscript to denote variables that are only associated with normal and anomaly points, respectively. Consider a toy dataset, where $\Delta\mathbf{x}_{t,n}^c$, $\Delta\mathbf{x}_{t,n}^p$, $\Delta\mathbf{x}_{t,a}^c$, and $\Delta\mathbf{x}_{t,a}^p$ have been defined:

$$\Delta\mathbf{x}_{t,n}^c \sim \mathcal{N}(0, I_D), \quad \Delta\mathbf{x}_{t,n}^p \sim \mathcal{N}(0, I_D), \quad \Delta\mathbf{x}_{t,a}^c \sim \mathcal{N}(0, I_D), \quad \Delta\mathbf{x}_{t,a}^p \sim \mathcal{N}(0, \alpha^2 I_D) \tag{2}$$

According to (1), we have

$$2N_n(t) = 2\frac{\|\Delta\mathbf{x}_{t,n}^c\|_2^2}{\|\Delta\mathbf{x}_{t,n}^c + \Delta\mathbf{x}_{t,n}^p\|_2^2} \sim \mathrm{F}(D, D) \tag{3}$$

$$(1 + \alpha^2)N_a(t) = (1 + \alpha^2)\frac{\|\Delta\mathbf{x}_{t,a}^c\|_2^2}{\|\Delta\mathbf{x}_{t,a}^c + \Delta\mathbf{x}_{t,a}^p\|_2^2} \sim \mathrm{F}(D, D) \tag{4}$$

where F is the F-distribution with $D$ and $D$ degrees of freedom. Fig. 2 illustrates the probability density function of $N_n(\cdot)$ and $N_a(\cdot)$ for different $D$ and $\alpha$. If $\alpha > 1$, then $N(\cdot)$ becomes an appropriate nominality score, since

$$\mathbb{P}(N(t) > \theta_N | y_t = 0) = \int_{2\theta_N}^{\infty} f(x; D, D)dx > \int_{(1+\alpha^2)\theta_N}^{\infty} f(x; D, D)dx = \mathbb{P}(N(t) > \theta_N | y_t = 1) \tag{5}$$

where $f(\cdot; D, D)$ is the probability density function of the F-distribution with degrees of freedom $D$ and $D$. Indeed, it is reasonable to assume that $\Delta\mathbf{x}_{t,a}^p$ has a larger variance than $\Delta\mathbf{x}_{t,n}^p$.

### 3.4 The Induced Anomaly Score

Having $N(\cdot)$ defined, we now propose a method for integrating any given $N(\cdot)$ and anomaly score $A(\cdot)$ to yield an *induced* anomaly score $\hat{A}(\cdot)$, and show some instances where the performance will improve over using $A(\cdot)$ or a smoothed $A(\cdot)$. Consider a dataset that contains subsequence anomalies. For two near time points $t$ and $\tau$, it is natural to assume that the possibility of $t$ being anomalous is affected by $\tau$. We quantify this effect as $A(t; \tau)$, which is the induced anomaly score at $t$ due to $\tau$. By summing over a range of $\tau$ around $t$, we get the induced anomaly score at $t$:

$$\hat{A}(t) \triangleq \sum_{\tau=\max(1, t-d)}^{\min(T, t+d)} A(t; \tau) \tag{6}$$

where $d$ is the induction length. Furthermore, we define $A(t; \tau)$ as a *gated* value of $A(\tau)$, which is controlled by the nominality score from $\tau$ to $t$:

$$A(t; \tau) \triangleq A(\tau) \prod_{k=\min(\tau+1, t)}^{\max(t - \mathbb{1}_{t=\tau}, \tau-1)} g_{\theta_N}(N(k)) = \begin{cases} A(\tau)g_{\theta_N}(N(\tau+1))...g_{\theta_N}(N(t)) & t > \tau \\ A(\tau) & t = \tau \\ A(\tau)g_{\theta_N}(N(\tau-1))...g_{\theta_N}(N(t)) & t < \tau \end{cases} \tag{7}$$

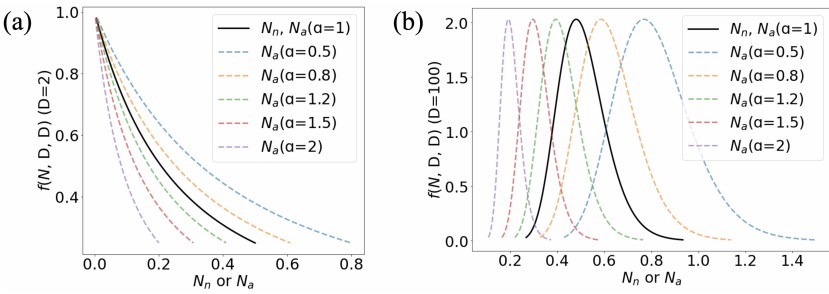

Figure 2: The probability density function for $N_n$ and $N_a$ of the toy dataset at (a) $D = 2$ and (b) $D = 100$.

where the gate function $g_{\theta_N}(N)$ is some transformation function of $N$ conditioned on a threshold $\theta_N$. A reasonable assumption is that $g_{\theta_N}(N)$ is a non-increasing function of $N$, i.e., $N > N'$ implies $g_{\theta_N}(N) \leq g_{\theta_N}(N')$. Indeed, if $N(k)$ is large, then time point $k$ is likely a normal point, and any two points $t_1, t_2$, where $t_1 < k < t_2$, are unlikely to be in the same anomaly subsequence, hence $A(t_1; t_2)$ and $A(t_2; t_1)$ should be small. Explicitly, we can use $\hat{A}(t) = \hat{A}(t; g_{\theta_N})$ and $A(t; \tau) = A(t; \tau, g_{\theta_N})$ to denote that these values are conditioned on $g_{\theta_N}$. Overall, $\hat{A}(\cdot)$ can be thought of as some (unnormalized) weighted smoothed value of $A(\cdot)$, where the weights are the product of $g_{\theta_N}(N(\cdot))$ across some range. We consider the following two cases:

**Claim 1**  Using a *soft* gate function,
$$g_{\theta_N}(N) \triangleq \max(0, 1 - \frac{N}{\theta_N}) \tag{8}$$

If there exists $\theta_1$ such that $N(t) \geq \theta_1$ for all normal points ($y_t = 0$), then $\text{F1}^*(\hat{A}(\cdot; g_{\theta_1}); \mathbf{y}) \geq \text{F1}^*(A(\cdot); \mathbf{y})$, i.e., the best F1 score using the induced anomaly score with $g_{\theta_1}$ as the gate function is greater or equal to the best F1 score using the original anomaly score.

**Proof 1**  For any normal time point $t_n$, we have
$$\hat{A}(t_n; g_{\theta_1}) = \sum_{\tau=\max(1,t_n-d)}^{\min(T,t_n+d)} A(t_n; \tau, g_{\theta_1}) = \sum_{\tau=\max(1,t_n-d)}^{\min(T,t_n+d)} A(\tau) \mathbb{1}_{t_n=\tau} = A(t_n) \tag{9}$$

The equality in the middle arises from the fact that $g_{\theta_1}(N(t_n)) = 0$, according to (8) and the assumptions. However, for any anomaly point $t_a$, we have
$$\hat{A}(t_a; g_{\theta_1}) = \sum_{\tau=\max(1,t_a-d)}^{\min(T,t_a+d)} A(t_a; \tau, g_{\theta_1}) \geq A(t_a) \tag{10}$$

This is because $N(t_a)$ might be lower than $\theta_1$, and hence $g_{\theta_1}(N(t_a)) \geq 0$. By potentially having a higher $\hat{A}(t)$ than $A(t)$ for anomaly points, we get a potentially higher $\text{F1}^*$.

Such a $\theta_1$ indeed exists in real applications (e.g. the minimum nominality score among all normal points $t_n$). However, targeting this value barely leads to any improvement for $\text{F1}^*$ in practice. This is because anomaly points are generally fewer than normal points, resulting in barely any anomaly points $t_a$ having $N(t_a) < \theta_1$. Nevertheless, we have shown that the soft gate function along with the induced anomaly score can provably yield equal or better $\text{F1}^*$ under some threshold.

**Claim 2**  Using a *hard* gate function,
$$g_{\theta_N}(N) \triangleq \mathbb{1}_{N<\theta_N} \tag{11}$$

If $d = 1$, and there exist two thresholds: (i) $\theta_2$ such that $N(t) < \theta_2$ for all anomaly time points ($y_t = 1$) (ii) $\theta_\infty = \infty$; then $\text{F1}^*(\hat{A}(\cdot; g_{\theta_2}); \mathbf{y}) \geq \text{F1}^*(\hat{A}(\cdot; g_{\theta_\infty}); \mathbf{y})$, i.e., the best F1 score using the induced anomaly score with $g_{\theta_2}$ as the gate function is greater or equal to the best F1 score using the induced anomaly score with $g_{\theta_\infty}$ as the gate function.

**Proof 2**  For any anomaly time point $t_a$, we have
$$\hat{A}(t_a; g_{\theta_2}) = \sum_{\tau=\max(1,t_a-1)}^{\min(T,t_a+1)} A(\tau) = A(t_a - 1)\mathbb{1}_{t_a>1} + A(t_a) + A(t_a + 1)\mathbb{1}_{t_a<T} \tag{12}$$

where the first equality arises from the fact that $g_{\theta_2}(N(t_a)) = 1$, according to (11) and the assumptions. For any normal time point $t_n$, we have
$$\hat{A}(t_n; g_{\theta_2}) = \sum_{\tau=\max(1,t_n-1)}^{\min(T,t_n+1)} A(t_n; \tau, g_{\theta_2}) \leq A(t_n - 1)\mathbb{1}_{t_n>1} + A(t_n) + A(t_n + 1)\mathbb{1}_{t_n<T} \tag{13}$$

since $N(t_n)$ might be greater than $\theta_2$ and hence $g_{\theta_2}(N(t_n)) \leq 1$. However, we have
$$\hat{A}(t; g_{\theta_\infty}) = \sum_{\tau=\max(1,t-1)}^{\min(T,t+1)} A(\tau) = A(t - 1)\mathbb{1}_{t>1} + A(t) + A(t + 1)\mathbb{1}_{t<T} \tag{14}$$

regardless of normal or anomaly points since $N(t) < \theta_\infty$ for any $t$. Therefore, since $\hat{A}(t_a; g_{\theta_2}) = \hat{A}(t_a; g_{\theta_\infty})$ and $\hat{A}(t_n; g_{\theta_2}) \leq \hat{A}(t_n; g_{\theta_\infty})$, we get a potentially higher F1* when using $\theta_2$ compared to using $\theta_\infty$.

$\hat{A}(\cdot; g_{\theta_\infty})$ can be viewed as the smoothed value (or shifted simple moving average) over $A(\cdot)$ with a period of $2d + 1$. This averaging method is common among other studies [11, 12, 32, 33]. Claim 2 implies that by conditioning on $N(\cdot)$ and calculating $\hat{A}(t)$, the performance can be improved over using a simple smoothing value of $A(\cdot)$. In practice, we can relax the constraint of $d$, and use other gated functions to yield a more flexible architecture. Note that the appropriateness of a nominality score is critical for this to work.

## 3.5 Point-based Reconstruction Models

Consider some model $\mathcal{M}_{pt}$ that reconstructs each time point $t$ point-wise: $\hat{\mathbf{x}}_{pt,t} \triangleq \mathcal{M}_{pt}(\mathbf{x}_t^0)$. Using $\mathcal{M}_{pt}$, a method for yielding the anomaly score is by using the point-based reconstruction mean-squared error, defined as $\mathbf{a}_{pt} = \{a_{pt,1}, ..., a_{pt,T}\}$, where $a_{pt,t} \triangleq \|\hat{\mathbf{x}}_{pt,t} - \mathbf{x}_t^0\|_2^2$. One concern for using this kind of anomaly score is that we are not taking into account any time-dependent relationships for deriving $\mathbf{a}_{pt}$, so simply using $\mathcal{M}_{pt}$ and $\mathbf{a}_{pt}$ can barely be classified as a time series anomaly detection approach. Surprisingly, however, we find that a simple realization of $\mathcal{M}_{pt}$ can already achieve impressive results for F1* (section 4.4).

Since $\mathcal{M}_{pt}$ learns to capture the distribution of all normal point data, we assume that $\hat{\mathbf{x}}_{pt,t} \in \mathcal{X}^*$ or is very close. Moreover, since $\hat{\mathbf{x}}_{pt,t}$ can offset point-anomalies, we assume $\mathbf{x}_t^c \approx \hat{\mathbf{x}}_{pt,t}$, and implement a point-based reconstruction model to calculate $\hat{\mathbf{x}}_{pt,t}$ in practice.

$$\mathbf{X}^c = \{\mathbf{x}_1^c, ..., \mathbf{x}_T^c\} \approx \hat{\mathbf{X}}^c = \{\hat{\mathbf{x}}_1^c, ..., \hat{\mathbf{x}}_T^c\}, \quad \hat{\mathbf{x}}_t^c = \hat{\mathbf{x}}_{pt,t} = \mathcal{M}_{pt}(\mathbf{x}_t^0) \tag{15}$$

$\mathcal{M}_{pt}$ can be any model that has the ability to reconstruct $\mathbf{X}^0$ point-wise. To our surprise, the best performance can be achieved by using a simple Performer-based autoencoder ([34]) that actually has the potential to discover temporal information. Despite having such a possibility, it only learned to reconstruct point-by-point during training. We demonstrated this fact by shuffling the input time points and observing that the result will be the same after reordering the output sequence. One possible explanation is that during training, it is a lot easier to individually reconstruct single time points than to find complex time-dependent relationships; and since the Performer-based autoencoder tries to optimize over a batch of time points, this reduces the effect of overfitting and allows the model to better generalize to unseen data. However, the exact reason for this remains an open question. For the rest of the study, we will use $\mathcal{M}_{pt}$ to refer to the Performer-based autoencoder model. Details for the architecture of $\mathcal{M}_{pt}$ are shown in Appendix B.1.

## 3.6 Sequence-based Reconstruction Models

Contrary to $\mathbf{x}_t^c$, $\mathbf{x}_t^*$ should not only be in $\mathcal{X}^*$ but also obey the time-dependent relationships. Therefore, it is necessary that the model ($\mathcal{M}_{seq}$) for approximating $\mathbf{x}_t^*$ takes a sequence of time points as input.

$$\mathbf{X}^* \approx \hat{\mathbf{X}}^* = \{\hat{\mathbf{x}}_1^*, ..., \hat{\mathbf{x}}_T^*\} \triangleq \mathcal{M}_{seq}(\mathbf{X}^0) \tag{16}$$

How close $\mathcal{M}_{seq}(\mathbf{X}^0)$ approximates $\mathbf{X}^*$ depends on the amount of training data and the model capacity. In practice, for computational reasons, only a section of $\mathbf{X}^0$ is input and reconstructed at a time. We found that given a subsequence $\mathbf{X}_{ab}^0 = \{\mathbf{x}_a^0, ..., \mathbf{x}_b^0\}$ as input for reconstruction, a model tends to simply reconstruct individual points and do not take temporal information into account (as discussed in section 3.5). Therefore, we use a Performer-based stacked encoder as $\mathcal{M}_{seq}$, which predicts the middle $\delta$ points from its surrounding $2\gamma$ points to force the learning of time-dependent relationships. We concatenate all the predicted time points output by $\mathcal{M}_{seq}$ to construct $\hat{\mathbf{X}}^*$. For the rest of the study, we will use $\mathcal{M}_{seq}$ to refer to the Performer-based stacked encoder model. Details for the architecture of $\mathcal{M}_{seq}$ are shown in Appendix B.2. Fig. 3 gives an illustration of the architecture for $\mathcal{M}_{pt}$, $\mathcal{M}_{seq}$, and the overall scheme. By obtaining $\hat{\mathbf{X}}^c$ and $\hat{\mathbf{X}}^*$, we can calculate $N(\cdot)$ and select some $A(\cdot)$ for calculating $\hat{A}(\cdot)$. The algorithm for evaluating a trained $\mathcal{M}_{pt}$ and $\mathcal{M}_{seq}$ using the soft gate function is shown in **Algorithm 1**.

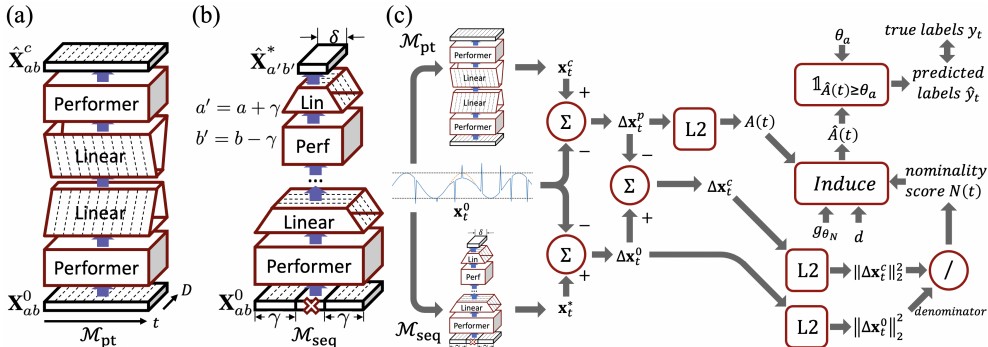

Figure 3: (a) Performer-based autoencoder $\mathcal{M}_{pt}$, (b) Performer-based stacked encoder $\mathcal{M}_{seq}$, and (c) main scheme for NPSR. GELUs are used as the activation function for each layer.

---

**Algorithm 1** NPSR F1* Evaluation (soft gate function)

---

**function** NPSR($\mathcal{M}_{pt}, \mathcal{M}_{seq}, \mathbf{X}^0 = \{\mathbf{x}_1^0, ..., \mathbf{x}_T^0\}, \mathbf{y} = \{y_1, ..., y_T\}, \theta_N, d$)

Construct $\hat{\mathbf{X}}^c = \{\hat{\mathbf{x}}_1^c, ..., \hat{\mathbf{x}}_T^c\}$ with $\hat{\mathbf{x}}_t^c \leftarrow \mathcal{M}_{pt}(\mathbf{x}_t^0)$         ▷ (15)

Construct $\hat{\mathbf{X}}^* = \{\hat{\mathbf{x}}_1^*, ..., \hat{\mathbf{x}}_T^*\} \leftarrow \mathcal{M}_{seq}(\mathbf{X}^0)$         ▷ (16)

Construct $A(\cdot)$ with $A(t) \leftarrow \|\hat{\mathbf{x}}_t^c - \mathbf{x}_t^0\|_2^2$         ▷ section 3.5

Construct $N(\cdot)$ with $N(t) \leftarrow \|\hat{\mathbf{x}}_t^c - \hat{\mathbf{x}}_t^*\|_2^2 / \|\mathbf{x}_t^0 - \hat{\mathbf{x}}_t^*\|_2^2$         ▷ (1)

Construct $g_{\theta_N}(N(\cdot))$ with $g_{\theta_N}(N(t)) \leftarrow \max(0, 1 - N(t)/\theta_N)$         ▷ (8)

Construct $A(\cdot; \cdot)$ with $A(t; \tau) \leftarrow A(\tau) \prod_{k=\min(\tau+1,t)}^{\max(t-\mathbb{1}_{t=\tau}, \tau-1)} g_{\theta_N}(N(k))$         ▷ (7)

Construct $\hat{A}(\cdot)$ with $\hat{A}(t) \leftarrow \sum_{\tau=\max(1,t-d)}^{\min(T,t+d)} A(t; \tau)$         ▷ (6)

return F1* $\leftarrow \max_{\theta_a} F1(\hat{\mathbf{y}}(\hat{A}(\cdot), \theta_a); \mathbf{y})$

---

# 4 Experiments

## 4.1 Datasets

We evaluate NPSR on the following datasets:

- **SWaT** (Secure Water Treatment) [35]: The SWaT dataset is collected over 11 days from a scaled-down water treatment testbed with 51 sensors. During the last 4 days, 41 anomalies were injected using diverse attack methods, while only normal data were generated during the first 7 days.

- **WADI** (WAter DIstribution testbed) [36]: The WADI dataset is acquired from a reduced city water distribution system with 123 sensors and actuators operating for 16 days. The first 14 days contain only normal data, while the remaining two days have 15 anomaly segments.

- **PSM** (Pooled Server Metrics) [37]: The PSM dataset is collected internally from multiple application server nodes at eBay. There are 13 weeks of training data and 8 weeks of testing data.

- **MSL** (Mars Science Laboratory) and **SMAP** (Soil Moisture Active Passive) [38, 13]: The MSL and SMAP datasets are public datasets collected by NASA, containing telemetry anomaly data derived from the Incident Surprise Anomaly (ISA) reports of spacecraft monitoring systems. The datasets have 55 and 25 dimensions respectively. The training set contains unlabeled anomalies.

- **SMD** (Server Machine Dataset) [9]: The SMD is collected from a large internet company, comprising 5 weeks of data from 28 server machines with 38 sensors. The first 5 days contain only normal data, and anomalies are injected intermittently for the last 5 days.

- **trimSyn** (Trimmed Synthetic Dataset) [24]: The original synthetic dataset was generated using trigonometric functions and Gaussian noises. We obtained the dataset from [39] and trimmed the test dataset such that only one segment of anomaly is present.

The statistics for the datasets are summarized in Table 1. For multi-entity datasets, Train#/Test# corresponds to the number of train/test time points summed over all entities, and the anomaly rate is calculated from the ratio between the sum of all anomaly points and sum of all test points.

Table 1: Datasets used in this study before preprocess.

| Dataset | Entities | Dims | Train # | Test # | Anomaly Rate (%) |
|---------|----------|------|---------|--------|------------------|
| SWaT | 1 | 51 | 495000 | 449919 | 12.14 |
| WADI | 1 | 123 | 1209601 | 172801 | 5.71 |
| PSM | 1 | 25 | 132481 | 87841 | 27.76 |
| MSL | 27 | 55 | 58317 | 73729 | 10.48 |
| SMAP | 55 | 25 | 140825 | 444035 | 12.83 |
| SMD | 28 | 38 | 708405 | 708420 | 4.16 |
| trimSyn | 1 | 35 | 10000 | 7680 | 2.34 |

## 4.2 Baselines

We evaluate the performance of NPSR against several deep learning algorithms and simple heuristics using F1*. Due to the exhaustive nature of optimizing for all datasets and algorithms, we follow a three-step approach to populate Table 2. Firstly, we reference values from the original paper, and if unavailable, we search for the highest reported values among other publications. Finally, if no reported values are found, we modify and run publicly available code. We cannot find any reported F1* of the PSM dataset using THOC or any publicly available code, hence leaving the value blank. For multi-entity datasets (MSL, SMAP, and SMD), we compare the performance using two methods - (1) combining all entities and training them together; (2) training each entity separately and averaging the results. Moreover, some literature attempt to find a threshold ($\theta_a$) and then calculate the performance conditioned on it [24, 40]. Since $\theta_a$ can simply be a one-value parameter, we assume that other studies have already optimized this value, and regard their reported F1 as F1*. More information on the sources of data can be found in Appendix E.

## 4.3 Main Results

In Table 2, we report the results for F1* on several datasets. Detailed preprocessing steps and training settings are reported in Appendix C. NPSR almost consistently outperforms other algorithms, only being slightly inferior to TranAD on the PSM dataset. NPSR makes use of $\mathcal{M}_{pt}$ to precisely capture point anomalies with low false-positive rates (given the best threshold). Moreover, it acquires the ability to detect contextual anomalies by incorporating $\mathcal{M}_{seq}$ through the calculation of $\hat{A}(\cdot)$, without compromising its capability of detecting point anomalies. We observe that recent studies do not necessarily have higher F1* scores than older ones. Interestingly, simple heuristics can even perform fairly well on F1*, with NPSR being the only algorithm consistently outperforming them.

Table 2: Best F1 score (F1*) results on several datasets, with bold text denoting the highest and underlined text denoting the second highest value. The deep learning methods are sorted with older methods at the top and newer ones at the bottom.

| Algorithm \ Dataset | SWaT | WADI | PSM | MSL | SMAP | SMD | trimSyn |
|---------------------|------|------|-----|-----|------|-----|---------|
| Simple Heuristic [11, 30, 31] | 0.789 | 0.353 | 0.509 | 0.239 | 0.229 | 0.494 | 0.093 |
| DAGMM [26] | 0.750 | 0.121 | 0.483 | 0.199 | 0.333 | 0.238 | 0.326 |
| LSTM-VAE [22] | 0.776 | 0.227 | 0.455 | 0.212 | 0.235 | 0.435 | 0.061 |
| MSCRED [24] | 0.757 | 0.046 | 0.556 | 0.250 | 0.170 | 0.382 | 0.340 |
| OmniAnomaly [9] | 0.782 | 0.223 | 0.452 | 0.207 | 0.227 | 0.474 | 0.314 |
| MAD-GAN [23] | 0.770 | 0.370 | 0.471 | 0.267 | 0.175 | 0.220 | 0.331 |
| MTAD-GAT [27] | 0.784 | 0.437 | 0.571 | 0.275 | 0.296 | 0.400 | 0.372 |
| USAD [28] | 0.792 | 0.233 | 0.479 | 0.211 | 0.228 | 0.426 | 0.326 |
| THOC [18] | 0.612 | 0.130 | - | 0.190 | 0.240 | 0.168 | - |
| UAE [11] | 0.453 | 0.354 | 0.427 | 0.451 | 0.390 | 0.435 | 0.094 |
| GDN [12] | 0.810 | 0.570 | 0.552 | 0.217 | 0.252 | 0.529 | 0.284 |
| GTA [41] | 0.761 | 0.531 | 0.542 | 0.218 | 0.231 | 0.351 | 0.256 |
| Anomaly Transformer [40] | 0.220 | 0.108 | 0.434 | 0.191 | 0.227 | 0.080 | 0.049 |
| TranAD [25] | 0.669 | 0.415 | **0.649** | 0.251 | 0.247 | 0.310 | 0.282 |
| NPSR (combined) | - | - | - | 0.261 | **0.511** | 0.227 | - |
| NPSR | **0.839** | **0.642** | 0.648 | **0.551** | 0.505 | **0.535** | **0.481** |

We suggest using these simple heuristic values as a strong baseline for future studies to compare against. In light of recent publications highlighting the limitations of using the point-adjusted F1 score ([11, 30, 31, 42]), and yet the large amount of work still using it, we also report the results using point-adjustment in Appendix D.

For multi-entity datasets, we observe that the standard method (training one point-based and sequence-based model per entity) outperforms the combined method for MSL and SMD datasets. This is not surprising, given that entity-to-entity variations might be large. However, for the SMAP dataset, we observe that the combined method performs better. We attribute such results to the fact that the SMAP spacecraft are routine, hence the resulting telemetry between entities can have similar underlying distributions. This contributes to additive learning from the increased training data [13].

### 4.4 Ablation Study

The ablation study conducted in this section sheds light on several aspects of the proposed method. In Table 3, we compare the performance of five methods that yield different anomaly scores (either $A(\cdot)$ or $\hat{A}(\cdot)$). For the first two methods, the reconstruction errors of $\mathcal{M}_{pt}$ and $\mathcal{M}_{seq}$ are used, respectively. Since $\mathcal{M}_{pt}$ mostly performs better than $\mathcal{M}_{seq}$, we use the point-based reconstruction error as $A(\cdot)$ to calculate $\hat{A}(\cdot)$ for the last three methods. The third method corresponds to an unnormalized simple smoothed value of $A(\cdot)$ (cf. section 3.4). The fourth and fifth methods use gate functions (11) and (8), respectively, but the same $\theta_N$ that corresponds to the 98.5 percentile of the nominality score from the training data ($N_{trn}$). This method for setting $\theta_N$ works well enough and can be applied across different datasets. Illustrations of the distribution of nominality scores for the SWaT and WADI datasets, along with the 98.5% threshold value are shown in Fig. 4. We can observe that the distributions are *close to appropriate* (cf. section 3.3). The results for the last three methods are averaged over induction lengths $d = 1, 2, 4, 8, 16, 32, 64, 128,$ and 256. Each entity is trained separately for multi-entity datasets and the results are pooled together.

Firstly, we observe that $\mathcal{M}_{pt}$ outperforms $\mathcal{M}_{seq}$ and achieves competitive results on its own, despite *not* modeling the time-dependent relationships. Secondly, by smoothing $A(\cdot)$ (third row), the AUC and F1* are increased for most datasets. This means smoothing is generally an effective method for improving performance. Moreover, our experiments show that the soft gate function along with an appropriate $\theta_N$ performed the best on average in terms of F1*. This suggests that the distribution of nominality scores is predominantly overlapped, and a soft gate function will be more appropriate to prevent excessive accumulation of anomaly scores on normal time points, reducing false-positives (cf. Appendix F). This method will also have a generally stable AUC and F1* (low $\sigma_d$) across a wide range of $d$. This makes sense - when time point $\tau$ is farther away from time point $t$, more gate function outputs are multiplied onto $A(\tau)$, hence $A(t, \tau) \to 0$. However, the results also suggest that the best choice of gate function and $\theta_N$ may depend on the specific dataset at hand.

### 4.5 Detection Trade-off Between Point and Contextual Anomalies

We elaborate on the trade-off between detecting point and contextual anomalies and relate them with the performance of $\mathcal{M}_{pt}$ and $\mathcal{M}_{seq}$. It may seem intuitive that finding temporal information in

Table 3: AUC and F1* for different methods and datasets, with bold text denoting the highest and underlined text denoting the second highest value. The mean ($\mu_d$) and standard deviation ($\sigma_d$) of the performance metrics evaluated across $d = 1, 2, 4, 8, 16, 32, 64, 128, 256$ are shown.

| Dataset | | SWaT | | WADI | | PSM | | MSL | | SMAP | | SMD | | trimSyn | |
|---|---|---|---|---|---|---|---|---|---|---|---|---|---|---|---|
| Method | | AUC | F1* | AUC | F1* | AUC | F1* | AUC | F1* | AUC | F1* | AUC | F1* | AUC | F1* |
| $\mathcal{M}_{pt}$ ($\|\hat{\mathbf{x}}_t^c - \mathbf{x}_t^0\|_2^2$) | | 0.908 | **0.839** | 0.819 | 0.629 | 0.790 | 0.626 | 0.640 | 0.366 | 0.647 | 0.329 | 0.820 | 0.485 | 0.721 | 0.100 |
| $\mathcal{M}_{seq}$ ($\|\hat{\mathbf{x}}_t^* - \mathbf{x}_t^0\|_2^2$) | | 0.899 | 0.755 | 0.843 | 0.559 | 0.766 | 0.576 | 0.621 | 0.351 | 0.611 | 0.292 | 0.820 | 0.482 | 0.832 | 0.345 |
| $\mathcal{M}_{pt}$ + Hard (11) | $\mu_d$ | **0.912** | 0.813 | 0.827 | 0.630 | 0.775 | 0.621 | 0.708 | 0.451 | **0.665** | **0.389** | 0.835 | 0.492 | 0.785 | 0.144 |
| ($\theta_N = \infty$) | $\sigma_d$ | 0.005 | 0.034 | 0.007 | 0.037 | 0.023 | 0.020 | 0.032 | 0.038 | 0.010 | 0.036 | 0.025 | 0.052 | 0.037 | 0.021 |
| $\mathcal{M}_{pt}$ + Hard (11) | $\mu_d$ | **0.912** | 0.820 | 0.844 | 0.625 | 0.779 | 0.624 | **0.718** | **0.467** | 0.659 | 0.386 | 0.833 | 0.495 | 0.791 | 0.292 |
| ($\theta_N = 98.5\% N_{trn}$) | $\sigma_d$ | 0.005 | 0.024 | 0.007 | 0.023 | 0.017 | 0.015 | 0.041 | 0.051 | 0.012 | 0.034 | 0.024 | 0.050 | 0.069 | 0.121 |
| $\mathcal{M}_{pt}$ + Soft (8) | $\mu_d$ | 0.909 | 0.837 | **0.856** | **0.639** | **0.804** | **0.636** | 0.698 | 0.465 | 0.656 | 0.388 | **0.840** | **0.525** | **0.862** | **0.434** |
| ($\theta_N = 98.5\% N_{trn}$) | $\sigma_d$ | 0.000 | 0.001 | 0.011 | 0.008 | 0.005 | 0.004 | 0.031 | 0.061 | 0.005 | 0.039 | 0.003 | 0.011 | 0.063 | 0.099 |

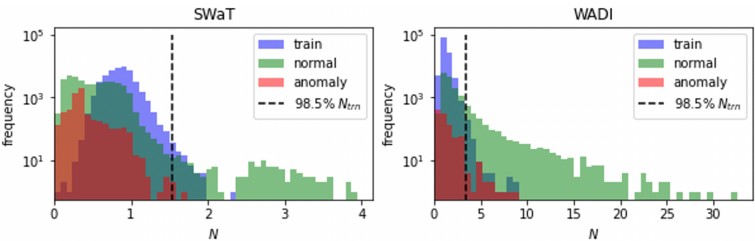

Figure 4: Histograms of the nominality scores for the SWaT and WADI dataset.

time series would lead to better performance. However, the modeling complexity increases with the number of time points being considered. This leads to the difficulty of focusing on the reconstruction of single time points. Fig. 5(a) shows $A(\cdot)$ calculated using either $\mathcal{M}_{pt}$ or $\mathcal{M}_{seq}$, and $\hat{A}(\cdot)$ calculated by NPSR using the WADI dataset. Given the fact that point $t$ is predicted anomalous if $A(t) \geq \theta_a$, we can see that $\mathcal{M}_{seq}$ has a higher false positive rate due to the spikes. In comparison, $\mathcal{M}_{pt}$ more accurately detects anomalies in a point-wise fashion. This highlights the superiority of $\mathcal{M}_{pt}$ over $\mathcal{M}_{seq}$ for this dataset. Furthermore, the induced anomaly score calculated by NPSR has very low false positive rates, and can sometimes even learn to identify anomalies that are not detected by $\mathcal{M}_{pt}$ or $\mathcal{M}_{seq}$ (Fig. 5(b), the anomaly subsequence at $t = 15200$ and the subsequence to its left, pointed by the black arrows). This suggests that $\hat{A}(\cdot)$ is superior to the original $A(\cdot)$ for this dataset. However, $\mathcal{M}_{pt}$ might not always perform superior to $\mathcal{M}_{seq}$. False negatives can be visualized between $t = 14800$ and $t = 14900$, where $\mathcal{M}_{pt}$ struggles to recognize the anomaly but $\mathcal{M}_{seq}$ effectively detects anomalous time-dependent relationships. This suggests that the anomaly segment contains relatively more contextual than point anomalies. Since the reconstruction error of $\mathcal{M}_{pt}$ is used as $A(\cdot)$, we lose the advantage of effectively utilizing the reconstruction error of $\mathcal{M}_{seq}$. This results in $\hat{A}(\cdot)$ not high enough to reach $\theta_a$ within this segment. An important future direction would be to explore how to appropriately select $A(\cdot)$ among multiple models (cf. Appendix F).

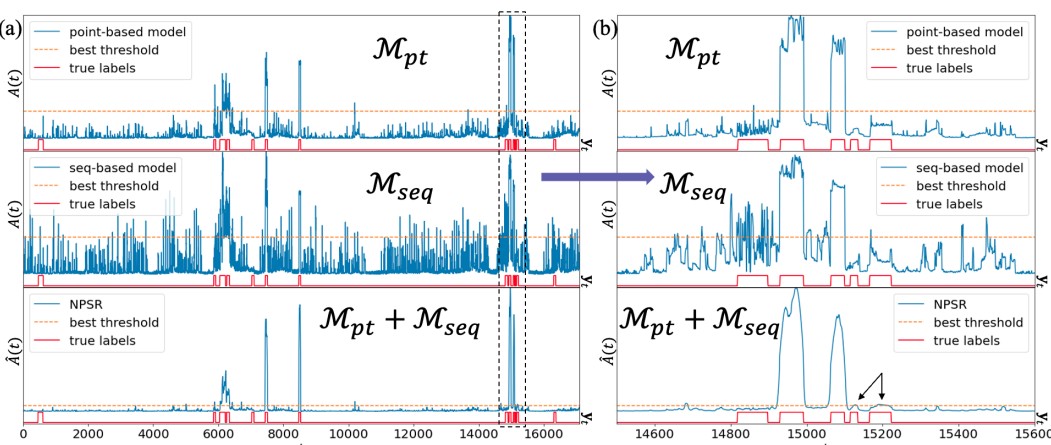

Figure 5: (a) Anomaly scores using $\mathcal{M}_{pt}$, $\mathcal{M}_{seq}$ and NPSR (soft gate function, $\theta_N = 99.85\% N_{trn}$, and $d = 16$), and the true labels of the WADI dataset. (b) Magnification for $t \in \{14500, ..., 15600\}$.

## 5 Conclusion

In conclusion, we introduce an improved framework for unsupervised time series anomaly detection. We specify the relationships between point and contextual anomalies and derive the nominality score and induced anomaly score to provide a theory-based algorithm with provable superiority. NPSR captures both point and contextual anomalies, resulting in a high combined precision and recall. Our results show that NPSR exhibits high performance, is widely applicable, and has a relatively straightforward training process. It has the potential to decrease labor needs for fault monitoring and correspondingly accelerates decision making and can also contribute to AI sustainability by preventing energy waste or system failure.

# 6  Acknowledgement

We thank Piyush Desai, Raphael Schutz, and Nikhil Deshmukh from Turntide Technologies for their helpful discussions and insights.

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

## A  The Best F1 Score

This section describes the method for calculating the best F1 score ($F1^*$) from a set of anomaly scores $\mathbf{a} = \{a_1, ..., a_T\}$ and a set of labels $\mathbf{y} = \{y_1, ..., y_T\}$. Firstly, given $\mathbf{a}$ and some arbitrary threshold $\theta_a$, we can calculate $\hat{\mathbf{y}} = \{\hat{y}_1, ...\hat{y}_T\}$, where $\hat{y}_t \triangleq \mathbb{1}_{a_t \geq \theta_a}$. Secondly, $\hat{\mathbf{y}}$ is used to calculate TP, FP, and FN, which corresponds to the sets of time points for true positives, false positives, and false negatives.

$$\text{TP} \triangleq \{t | \hat{y}_t = 1, y_t = 1\}, \quad \text{FP} \triangleq \{t | \hat{y}_t = 1, y_t = 0\}, \quad \text{FN} \triangleq \{t | \hat{y}_t = 0, y_t = 1\} \tag{17}$$

Thirdly, we calculate the precision (P) and recall (R), and then calculate the F1 score, which is the harmonic mean between R and P.

$$\text{F1} \triangleq \frac{2\text{PR}}{\text{P} + \text{R}}, \quad \text{P} \triangleq \frac{\text{n(TP)}}{\text{n(TP)} + \text{n(FP)}}, \quad \text{R} \triangleq \frac{\text{n(TP)}}{\text{n(TP)} + \text{n(FN)}} \tag{18}$$

Finally, we calculate $F1^*$ by using the threshold that yields the highest F1.

$$\text{F1}^*(\mathbf{a}; \mathbf{y}) \triangleq \max_{\theta_a} \text{F1}(\hat{\mathbf{y}}(\mathbf{a}, \theta_a); \mathbf{y}) \tag{19}$$

## B  Model Architecture

Performers (an improved variant of Transformers) are competitive in terms of execution speed compared with other Transformer variants [34, 43], hence we use them as the basic building block of our models. For both $\mathcal{M}_{pt}$ and $\mathcal{M}_{seq}$, we use a linear layer with input and output dimensions equal to $D$ as the token embedding layer, a fixed positional embedding layer at the beginning, a feature redraw interval of 1, and a tanh activation function immediately before the output. GELUs [44] are used as the activation layer for all linear layers. We do not change any other predefined activation layer inside Performers.

### B.1  Performer-based autoencoder

For the Performer-based autoencoder $\mathcal{M}_{pt}$, the input with the shape $(batch, W, D)$ is passed through a plain Performer with $N_{perf}$ layers after the positional embedding step, where $W$ represents the input window size. This is followed by a linear encoding layer that transforms the dimensionality from $D$ to $D_{lat}$, resulting in the shape $(batch, W, D_{lat})$ for the latent variables. In the case of $\mathcal{M}_{pt}$, there is no compression along the time domain. The latent variables then pass through another linear decoding layer that transforms the dimensionality from $D_{lat}$ back to $D$, followed by another Performer with $N_{perf}$ layers.

### B.2  Performer-based stacked encoder

For the Performer-based stacked encoder $\mathcal{M}_{seq}$, the input with shape $(batch, W_0, D)$ is passed through a plain Performer with one layer after the positional embedding step, followed by a linear encoding layer that transforms the window size from $W_0$ to $W_1$, where $W_0 = 2\gamma$ (cf. section 3.6). It should be noted that unlike $\mathcal{M}_{pt}$, compression is done along the time domain for $\mathcal{M}_{seq}$. The one-layer Performer and linear layer are stacked $N_{enc}$ times, where in the $i$-th stack, the window size is compressed from $W_{i-1}$ to $W_i$. $W_{N_{enc}}$ is equal to the target output window size $\delta$. For both $\mathcal{M}_{pt}$ and $\mathcal{M}_{seq}$, we optimize $N_{perf}$, $D_{lat}$, $W$, $N_{enc}$, $W_i$ for $i \in \{0, ..., N_{enc}\}$, and $\delta$ for the best performance. Note that $\mathcal{M}_{seq}$ isn't capable of reconstructing the first and last $\gamma$ time points due to its architecture, hence we discard the first and last $\gamma$ points reconstructed by $\mathcal{M}_{pt}$ so that rest of the time points have exactly two reconstructed values corresponding to using $\mathcal{M}_{seq}$ and $\mathcal{M}_{pt}$, respectively.

## C Data Preprocessing and Training Details

Table 4 shows the hyperparameters used for implementing NPSR on the experimented datasets. To ensure fair comparison, the same preprocessing method is applied to all algorithms for the same dataset. The search for hyperparameters is done manually, starting from some reasonable value (e.g. a learning rate of $10^{-4}$). The authors believe that there is still room for improvement by fine-tuning these hyperparameters. To speed up training, we load all training inputs and outputs, and testing inputs onto the GPU before training. We use a local GPU, which can be either GeForce RTX 3070 (8GB), 3080 (12GB) or 3090 (24GB). For an individual experiment using a single dataset and training method, the training time ranges from approximately 2 minutes to 12 hours. For single-entity datasets and multi-entity datasets that use the combined training method, we run the experiments for at least 3 times and confirm that the results are stable given different random seeds. For multi-entity datasets with entities trained individually, the results are averaged across all entities. Generally, datasets with single entities train faster than those with multiple entities. There are some additional remarks regarding the preprocessing of the datasets.

For SWaT, we use `SWaT_Dataset_Attack_v0.csv` and `SWaT_Dataset_Normal_v1.csv` from the folder `SWaT.A1 & A2_Dec 2015` (manually converted from `*.xlsx`). We corrected some original flaws in the dataset (e.g. redundant blank spaces in some labels), and set the 5th and 10th columns to all 0. For WADI, we use the 2017 year dataset. Columns with excessive NaNs (more than half of the entire length) are deleted. Other NaNs are forward-filled. After deleting all the columns with excessive NaNs, the 86th column is further set to all 0. For PSM, we forward-fill all NaNs. For MSL and SMD, two additional blank channels are added to make the number of channels divisible by the number of heads. For trimSyn, we separated the dataset into training ($t \in \{0, ..., 9999\}$) and testing ($t \in \{10000, ..., 19999\}$) data (cf. [24]). For the testing data, we extracted segments within the time intervals $t \in \{10000, ..., 11719\} \cup \{11900, ..., 12849\} \cup \{14630, ..., 17699\} \cup \{17880, ..., 18529\} \cup \{18710, ..., 19999\}$ and concatenated them to form a single-entity test dataset. We also inserted additional segment IDs using one-hot encoding to enable segment identification. This process resulted in a test dataset with a single anomaly segment ($t \in \{12670, ..., 12849\}$) and a corresponding anomaly rate of $2.34\%$. In the case of the training data, we duplicated the original segment five times, concatenated them, and added the necessary segment IDs.

Table 4: Implementation details. (c) stands for the combined training method (cf. section 4.2)

| Parameter \ Dataset | SWaT | WADI | PSM | MSL | MSL (c) | SMAP | SMAP (c) | SMD | SMD (c) | trimSyn |
|---|---|---|---|---|---|---|---|---|---|---|
| | | | | Preprocess | | | | | | |
| Downsample | 10 | 10 | 10 | 1 | 1 | 1 | 1 | 2 | 2 | 1 |
| Normalization | Minmax | Minmax | Minmax | Minmax | Minmax | Minmax | Minmax | Minmax | Minmax | Minmax |
| Stride | 10 | 10 | 10 | 10 | 10 | 10 | 10 | 10 | 10 | 10 |
| $W$ (for $\mathcal{M}_{pt}$) | 100 | 100 | 100 | 100 | 100 | 50 | 50 | 50 | 50 | 50 |
| $W_0$ (for $\mathcal{M}_{seq}$) | 100 | 100 | 100 | 50 | 50 | 50 | 50 | 50 | 50 | 50 |
| $\delta$ | 20 | 20 | 20 | 6 | 6 | 6 | 6 | 6 | 6 | 6 |
| | | | | Model architecture | | | | | | |
| # of heads | 9 | 14 | 5 | 11 | 12 | 5 | 10 | 8 | 11 | 5 |
| $D_{lat}$ | 10 | 10 | 10 | 10 | 10 | 10 | 10 | 10 | 10 | 4 |
| ff_mult | 4 | 4 | 4 | 4 | 4 | 4 | 4 | 4 | 4 | 4 |
| $N_{perf}$ | 4 | 4 | 4 | 4 | 4 | 4 | 4 | 4 | 4 | 4 |
| $N_{enc}$ | 8 | 8 | 8 | 8 | 8 | 8 | 8 | 8 | 8 | 4 |
| | | | | Induced anomaly score | | | | | | |
| Gate function | soft | soft | soft | soft | soft | soft | soft | soft | soft | soft |
| $d$ | 16 | 16 | 64 | 128 | 32 | 64 | 32 | 16 | 256 | 256 |
| Ratio of $N_{trn}$ for $\theta_N$ | 99.85% | 99.85% | 99.85% | 99.85% | 97.5% | 99.85% | 99.85% | 99.85% | 99.85% | 99.85% |
| | | | | Training | | | | | | |
| Learn rate | $10^{-4}$ | $10^{-4}$ | $10^{-4}$ | $10^{-4}$ | $10^{-4}$ | $10^{-4}$ | $10^{-4}$ | $10^{-4}$ | $10^{-4}$ | $10^{-4}$ |
| Optimizer | Adam | Adam | Adam | Adam | Adam | Adam | Adam | Adam | Adam | Adam |
| Batch size | 64 | 64 | 64 | 64 | 64 | 64 | 64 | 64 | 64 | 64 |
| Training epochs | 100 | 100 | 100 | 100 | 100 | 100 | 100 | 100 | 100 | 25 |

## D  The Point-adjusted Best F1 Score

Analogous to $F1^*$, the point-adjusted best F1 score ($F1^*_{PA}$) corresponds to $F1^*$ calculated after point-adjustment. Table 5 shows the $F1^*$ and $F1^*_{PA}$ of different algorithms, including NPSR, applied to several datasets. We did not show the results for the trimSyn dataset with point-adjustment. However, by applying NPSR and the same setting as without using point-adjust, we can achieve $F1^*_{PA} = 1$ within the first few epochs. This indicates the point with the highest anomaly score lies within the single anomaly segment.

The results suggest that $F1^*_{PA}$ may not be reliable - on the SWaT, WADI, PSM, and MSL datasets, simple heuristic approaches (e.g. using the mean squared value of an input time point as the anomaly score) outperform all deep learning methods when evaluated using $F1^*_{PA}$. Moreover, optimizing on $F1^*_{PA}$ does not necessarily guarantee a higher $F1^*$. NPSR is optimized on $F1^*$ by tuning the algorithm-specific parameters (e.g. $d$) and general parameters. To additionally optimize on $F1^*_{PA}$, we simply added *spikes* to the induced anomaly score ($\hat{A}_{spike}(\cdot)$) with value $\infty$ for some fixed interval $s$. The results show that $\hat{A}_{spike}(\cdot)$ can also achieve competitive $F1^*_{PA}$ values.

Table 5: Point-adjusted best F1 score ($F1^*_{PA}$) and best F1 score ($F1^*$) results on several datasets, with bold text denoting the highest and underlined text denoting the second highest value. The deep learning methods are sorted with older methods at the top and newer ones at the bottom.

| Dataset | SWaT | | WADI | | PSM | | MSL | | SMAP | | SMD | |
|---|---|---|---|---|---|---|---|---|---|---|---|---|
| Metric | $F1^*_{PA}$ | $F1^*$ | $F1^*_{PA}$ | $F1^*$ | $F1^*_{PA}$ | $F1^*$ | $F1^*_{PA}$ | $F1^*$ | $F1^*_{PA}$ | $F1^*$ | $F1^*_{PA}$ | $F1^*$ |
| Simple Heuristics [11, 30, 31] | **0.969** | 0.789 | **0.965** | 0.353 | **0.985** | 0.509 | **0.965** | 0.239 | 0.961 | 0.229 | 0.934 | 0.494 |
| DAGMM [26] | 0.853 | 0.750 | 0.209 | 0.121 | 0.761 | 0.483 | 0.701 | 0.199 | 0.712 | 0.333 | 0.723 | 0.238 |
| LSTM-VAE [22] | 0.805 | 0.776 | 0.380 | 0.227 | 0.809 | 0.455 | 0.854 | 0.212 | 0.756 | 0.235 | 0.808 | 0.435 |
| MSCRED [24] | 0.807 | 0.757 | 0.374 | 0.046 | 0.626 | 0.556 | 0.936 | 0.250 | 0.866 | 0.170 | 0.841 | 0.382 |
| OmniAnomaly [9] | 0.866 | 0.782 | 0.417 | 0.223 | 0.664 | 0.452 | 0.901 | 0.207 | 0.854 | 0.227 | 0.962 | 0.474 |
| MAD-GAN [23] | 0.815 | 0.770 | 0.556 | 0.370 | 0.658 | 0.471 | 0.917 | 0.267 | 0.865 | 0.175 | 0.915 | 0.220 |
| MTAD-GAT [27] | 0.860 | 0.784 | 0.602 | 0.437 | 0.780 | 0.571 | 0.908 | 0.275 | 0.901 | 0.296 | 0.908 | 0.400 |
| USAD [28] | 0.846 | 0.792 | 0.430 | 0.233 | 0.725 | 0.479 | 0.911 | 0.211 | 0.819 | 0.228 | 0.946 | 0.426 |
| THOC [18] | 0.881 | 0.612 | 0.506 | 0.130 | 0.895 | - | 0.937 | 0.190 | 0.952 | 0.240 | 0.541 | 0.168 |
| UAE [11] | 0.869 | 0.453 | 0.957 | 0.354 | 0.936 | 0.427 | 0.920 | 0.451 | 0.896 | 0.390 | **0.972** | 0.435 |
| GDN [12] | 0.935 | 0.810 | 0.855 | 0.570 | 0.923 | 0.552 | 0.903 | 0.217 | 0.708 | 0.252 | 0.716 | 0.529 |
| GTA [41] | 0.910 | 0.761 | 0.84 | 0.531 | 0.855 | 0.542 | 0.911 | 0.218 | 0.904 | 0.231 | 0.919 | 0.351 |
| Anomaly Transformer [40] | 0.941 | 0.019 | 0.714 | 0.015 | 0.979 | 0.022 | 0.936 | 0.021 | 0.967 | 0.019 | 0.923 | 0.021 |
| TranAD [25] | 0.815 | 0.669 | 0.495 | 0.415 | 0.882 | **0.649** | 0.949 | 0.251 | 0.892 | 0.247 | 0.961 | 0.310 |
| NPSR (combined) | - | - | - | - | - | - | 0.960 | 0.261 | **0.978** | **0.511** | 0.850 | 0.252 |
| NPSR | 0.953 | **0.839** | 0.938 | **0.642** | 0.957 | 0.648 | - | **0.551** | - | 0.437 | - | **0.535** |

# E  Source of Data

Table 6 shows the data sources used to produce Table 5, as well as the sources for section 4.3. The reference number is followed by a number between 1 and 3, where 1 indicates that the data comes from the original work, 2 indicates that the data comes from reproduced values of another literature, and 3 indicates that we have reproduced the values using public repositories..

Table 6: Data sources for algorithms and datasets. **Reproduced by using the squared value of channel 1 as $A(\cdot)$. †Reproduced by using the squared value of channel 9 as $A(\cdot)$.

| Dataset | SWaT | | WADI | | PSM | | MSL | | SMAP | | SMD | | trimSyn |
|---|---|---|---|---|---|---|---|---|---|---|---|---|---|
| Metric | $F1^*_{PA}$ | $F1^*$ | $F1^*_{PA}$ | $F1^*$ | $F1^*_{PA}$ | $F1^*$ | $F1^*_{PA}$ | $F1^*$ | $F1^*_{PA}$ | $F1^*$ | $F1^*_{PA}$ | $F1^*$ | $F1^*$ |
| Simple Heuristic | [30] - 1 | [30] - 1 | [30] - 1 | [30] - 1 | [31] - 1 | ** | [31] - 1 | [30] - 1 | [31] - 1 | [30] - 1 | [31] - 1 | [30] - 1 | † |
| DAGMM | [30] - 2 | [25] - 3 | [30] - 2 | [30] - 2 | [25] - 3 | [25] - 3 | [30] - 2 | [30] - 2 | [30] - 2 | [30] - 2 | [30] - 2 | [30] - 2 | [25] - 3 |
| LSTM-VAE | [28] - 2 | [28] - 2 | [28] - 2 | [28] - 2 | [40] - 2 | [45] - 3 | [28] - 2 | [30] - 2 | [28] - 2 | [30] - 2 | [28] - 2 | [30] - 2 | [45] - 3 |
| MSCRED | [25] - 2 | [25] - 3 | [25] - 2 | [25] - 3 | [25] - 3 | [25] - 3 | [25] - 2 | [25] - 3 | [25] - 2 | [25] - 3 | [25] - 2 | [25] - 3 | [25] - 3 |
| OmniAnomaly | [30] - 2 | [30] - 2 | [30] - 2 | [30] - 2 | [25] - 3 | [25] - 3 | [9] - 1 | [30] - 2 | [9] - 1 | [30] - 2 | [9] - 1 | [30] - 2 | [25] - 3 |
| MAD-GAN | [25] - 3 | [23] - 1 | [25] - 3 | [23] - 1 | [25] - 3 | [25] - 3 | [25] - 2 | [25] - 3 | [25] - 2 | [25] - 3 | [25] - 2 | [25] - 3 | [25] - 3 |
| MTAD-GAT | [46] - 2 | [27] - 3 | [46] - 2 | [27] - 3 | [27] - 3 | [27] - 3 | [27] - 1 | [27] - 3 | [27] - 1 | [27] - 3 | [46] - 2 | [27] - 3 | [25] - 3 |
| USAD | [28] - 1 | [28] - 1 | [28] - 1 | [28] - 1 | [25] - 3 | [25] - 3 | [28] - 1 | [30] - 2 | [28] - 1 | [30] - 2 | [28] - 1 | [30] - 2 | [25] - 3 |
| THOC | [18] - 1 | [30] - 2 | [30] - 2 | [30] - 2 | [40] - 2 | - | [18] - 1 | [30] - 2 | [18] - 1 | [30] - 2 | [30] - 2 | [30] - 2 | - |
| UAE | [11] - 1 | [11] - 1 | [11] - 1 | [11] - 1 | [11] - 3 | [11] - 3 | [11] - 1 | [11] - 1 | [11] - 1 | [11] - 1 | [11] - 1 | [11] - 1 | [11] - 3 |
| GDN | [30] - 2 | [12] - 1 | [30] - 2 | [12] - 1 | [12] - 3 | [12] - 3 | [30] - 2 | [30] - 2 | [30] - 2 | [30] - 2 | [30] - 2 | [30] - 2 | [25] - 3 |
| GTA | [41] - 1 | [41] - 3 | [41] - 1 | [41] - 3 | [41] - 3 | [41] - 3 | [41] - 1 | [41] - 3 | [41] - 1 | [41] - 3 | [41] - 3 | [41] - 3 | [41] - 3 |
| AnomalyTransformer | [40] - 1 | [40] - 3 | [40] - 3 | [40] - 3 | [40] - 1 | [40] - 3 | [40] - 1 | [40] - 3 | [40] - 1 | [40] - 3 | [40] - 1 | [40] - 3 | [40] - 3 |
| TranAD | [25] - 1 | [25] - 3 | [25] - 1 | [25] - 3 | [25] - 3 | [25] - 3 | [25] - 1 | [25] - 3 | [25] - 1 | [25] - 3 | [25] - 1 | [25] - 3 | [25] - 3 |

# F  Model and Parameter Selection Heuristics

We believe that the most important idea in model selection is to achieve a balance between effectively utilizing both point and sequence-based models and carefully selecting the appropriate parameters. As an extreme example, we optimize our algorithm on the Mackey-Glass anomaly benchmark (MGAB) [47], which is a univariate time series dataset, and find that sequence-based models can significantly outperform point-based models when using their reconstruction errors as $A(\cdot)$. As in Fig. 6, we see that $\mathcal{M}_{seq}$ can correctly identify the anomalies, whereas $\mathcal{M}_{pt}$ did not learn at all. This is reasonable, as point-based models only consider a single time point. In general, the lower the dimensionality of the dataset, the harder it gets for a point-based model to learn statistically meaningful representations. Moreover, looking at the high-dimensional datasets reported in Table 3, the induced anomaly score remains important for some datasets: For the MSL dataset, $F1^*$ improves 0.1 compared to only using point-based reconstruction. In Fig. 7b, when using a soft gate function, $F1^*$ improves 0.047 compared to using the point-based anomaly score. Since the amount of improvement depends on the statistical structure of the test data, it is still useful to consider using both $\mathcal{M}_{seq}$ and $\mathcal{M}_{pt}$ in general.

The difficulty in choosing the parameters for unsupervised time series anomaly detection stems from the absence of anomalies in the training dataset. The selection of soft or hard gate functions, the value for $d$, and the ratio for $\theta_N$ largely depend on how we presume the anomaly will occur based on domain knowledge. For instance, $d$ controls the distance that anomaly scores may propagate. This value should be higher if we presume the average anomaly length is long and vice versa. If

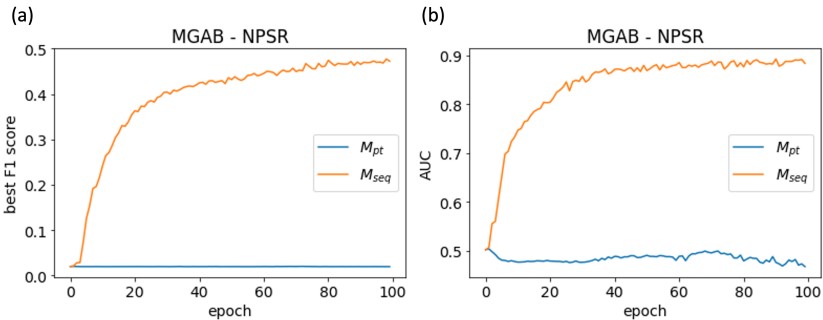

Figure 6: (a) best F1 scores and (b) AUCs using point and sequence-based models as the anomaly score on the Mackey-Glass anomaly benchmark.

the anomaly is expected to occur abruptly and significantly, there would be a clear gap between the distribution of nominality scores for normal and anomaly data (Fig. 8a). In this case, a hard gate function should be chosen as it allows anomaly scores to propagate through time points without reduction, as long as an anomaly time point has a nominality score lower than $\theta_N$. Conversely, if the anomaly occurs progressively, the distribution of nominality scores is likely to overlap (Fig. 8b). Here, a soft gate function will be more appropriate to prevent excessive accumulation of anomaly scores on normal time points, reducing false-positives. A dataset could contain both abrupt and progressive anomalies. However, based on Table 3, it is evident that using a soft gate function generally yields better performance compared to a hard gate function. This suggests that the distribution of nominality scores is predominantly overlapped, which is also evident in Fig. 4 and Fig. 7a.

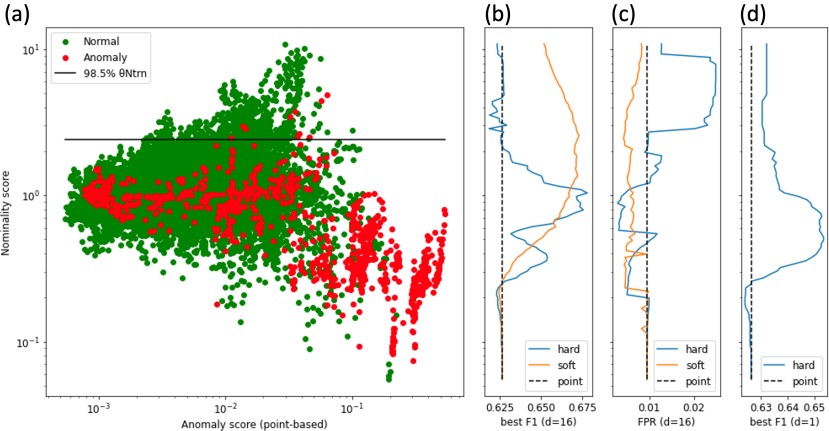

Figure 7: Training results on the WADI dataset. Nominality score vs (a) anomaly score from point-based reconstruction, (b) best F1 score ($d = 16$), (c) false positive rate ($d = 16$), and (d) best F1 score (d = 1) using different $\theta_N$.

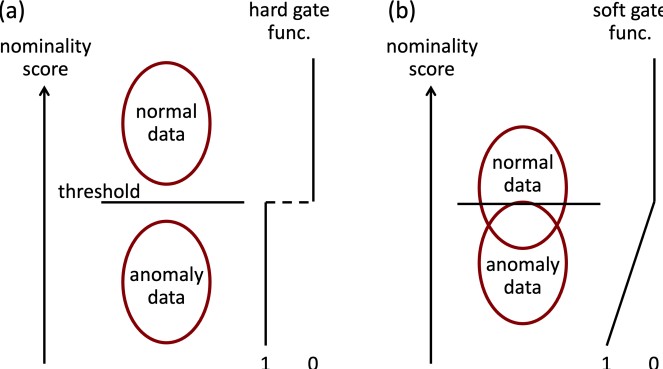

Figure 8: Illustration of different distributional relationships between the nominality scores of normal and anomaly data. (a) No overlap with a paired threshold and hard gate function. (b) Overlapped with threshold and soft gate function.

## G   Broader Impacts

The detection of anomalies in time series data can minimize downtime and avert financial losses. Utilizing real-time monitoring of system conditions, anomaly detection techniques for time series data can automatically detect deviations from the expected system behavior, thereby avoiding potential risks and financial harm. This has the potential to reduce the need for manual monitoring of faults and to expedite decision-making processes. Additionally, it can promote the sustainability of AI by preventing energy wastage and system malfunction.

# H   Limitations

Despite exhibiting competitive performance against other models, the proposed NPSR algorithm has a few limitations. The point-based model used in training does not incorporate temporal information, which makes it challenging to effectively reconstruct low-dimensional datasets. This issue is particularly challenging for univariate time series since raw inputs would not work for point-based models. One possible solution to this problem is to increase dimensionality by aggregating multiple time points. However, the effectiveness of this approach is yet to be confirmed.

Another limitation of NPSR is the absence of an automatic threshold ($\theta_a$)-finding method, which makes it difficult to determine a suitable threshold when deploying the model. To address this issue, one can define a target false positive rate and estimate the threshold that achieves this target rate using the validation set since only normal data is needed. Similarly, estimating the optimal values for $\theta_N$, $d$, and selecting the model used for calculating $A(\cdot)$ will be an important future work.

Within Fig. 5b of the main text, a false negative instance was identified in the temporal range between $t = 14800$ and $t = 14900$ when employing the induced anomaly score. According to the WADI dataset, this anomaly spans approximately 14.26 minutes and is characterized as "Damage 1 MV 001 and raw water pump to drain Elevated Reservoir tank." Notably, our analysis suggests that when assessing individual time points, the model $\mathcal{M}_{pt}$ encounters difficulty in recognizing this anomaly. Conversely, the model $\mathcal{M}_{seq}$ excels in identifying time-dependent relationships, making it more effective in capturing such contextual anomalies. The observed disparity in anomaly detection implies that this section comprises a relatively higher proportion of contextual anomalies than point anomalies. Consequently, when using $\mathcal{M}_{seq}$, we achieve a higher anomaly score. However, our current approach utilizes the reconstruction error of $\mathcal{M}_{pt}$ as the basis for the anomaly score calculation, thus neglecting the effectiveness of the reconstruction error generated by $\mathcal{M}_{seq}$. Consequently, the induced anomaly score fails to surpass the predefined threshold. In light of these findings, an essential avenue for future research is to investigate methods for selecting the model to be used in the computation of $A(\cdot)$. This undertaking holds promise for enhancing the overall performance and accuracy of anomaly detection in time series data.

