# OpenReview forum: "Nominality Score Conditioned Time Series Anomaly Detection by Point/Sequential Reconstruction"
_NeurIPS.cc/2023/Conference — NeurIPS 2023 poster_

### Official Review · Reviewer_UPqe · 2023-07-02

**Soundness:** 2 fair
**Presentation:** 1 poor
**Contribution:** 2 fair
**Rating:** 5
**Confidence:** 4

**Summary:**

TSAD techniques are targeted towards detecting either point or contextual anomalies, but often struggle to adequately capture both simultaneously. In this work, the authors propose a novel reconstruction-based AD technique that introduces the notion of a nominality score and a subsequent induced anomaly score. The method achieves a better trade-off between detecting point and contextual anomalies than competing techniques, and outperforms them over a variety of benchmark multivariate datasets.

**Strengths:**

- The problem statement is interesting and relevant, and I appreciate the use of visualisations such as in Figure 1, that convey the difficulty of the task.
- I appreciated that the model has several components that could be tweaked, such as the choice of soft or hard gate function, and the ablation study does a good job in highlighting their individual contributions. I would however have liked to see more discussion or synthetic examples that demonstrate whether there are certain properties of the monitored data that can guide the selection (beyond empirical comparison).

**Weaknesses:**

- I found the presentation and clarity to be quite poor overall, as a result of which the contributions are sometimes difficult to follow properly.  Section 2.2 introduces a lot of similar notations that are difficult to keep track of – a figure could be very helpful here for conveying the differences between the various time series and deviations.
- The authors rely on results reported in earlier papers as performance measures for several competing techniques. While I appreciate that it may be time-consuming to re-implement and re-run all experiments, I do worry about potential inconsistencies in the experimental set-up that may involuntarily skew the comparison. The authors mention how if the results from the original paper are unavailable, *“we search for the highest reported values among other publications.”* I can see this as introducing several inconsistencies, especially since the source of the result isn’t reported alongside the performance figures presented in the table.
- I would have liked there to be a dedicated *Related Work* section in the main paper in order to better understand how the work fits alongside other reconstruction-based techniques.

**Questions:**

I would appreciate if the authors could respond to the concerns listed in the *Weaknesses* section, and either defend their approach, or indicate how they can improve on the existing submission.

Although I find the problem investigated in the paper to be interesting and well-motivated, I don’t think that the submission is ready for publication in its current state, and believe it would strongly benefit from another revision/round of reviews.

---

> ### Author Rebuttal · Authors · 2023-08-09
>
> We extend our sincere gratitude to Reviewer *UPqe* for their detailed review and valuable insights. We will address the raised points in the revised version accordingly.
>
> #### 1. More discussion or synthetic examples that demonstrate whether there are certain properties of the monitored data that can guide the selection (beyond empirical comparison).
>
> > The difficulty in choosing these parameters stems from the absence of anomalies in the training dataset. The selection of soft or hard gate functions, the value for $d$, and the ratio for $\theta_N$ largely depend on how we presume the anomaly will occur based on domain knowledge.
> For example, if the anomaly is expected to occur abruptly and significantly, there would be a clear gap between the distribution of nominality scores for normal and anomaly data (**rebuttal Figure 2 (a)**). In this case, a hard gate function should be chosen as it allows anomaly scores to propagate through time points without reduction, as long as an anomaly time point has a nominality score lower than $\theta_N$. Conversely, if the anomaly occurs progressively, the distribution of nominality scores is likely to overlap (**rebuttal Figure 2 (b)**). Here, a soft gate function will be more appropriate to prevent excessive accumulation of anomaly scores on normal time points, reducing false-positives.
> A dataset could contain both abrupt and progressive anomalies. However, based on (main text) **Table 3**, it is evident that using a soft gate function generally yields better performance compared to a hard gate function. This suggests that the distribution of nominality scores is predominantly overlapped, which is also evident in (main text) **Figure 3** and **rebuttal Figure 3**.
> Another heuristic for selecting whether to use the point or seq-based reconstruction error as $A(\cdot)$ is by considering the dataset's dimensionality. In an experiment conducted on a univariate dataset, we demonstrated that a sequence-based model outperforms a point-based model significantly (**rebuttal Figure 1**). This is because when dimensionality reduces, extracting meaningful statistical structure from a single time point becomes more challenging.
> We will include relevant additional discussions in the revision.
>
> #### 2. Section 2.2 introduces a lot of similar notations that are difficult to keep track of – a figure could be very helpful here for conveying the differences between the various time series and deviations.
>
> > (main text) **Figure 2 (a)** illustrates the relationships for the introduced time series notations, including **x**$^0_t$, **x**$^p_t$, **x**$^c_t$, Δ**x**$^0_t$, … etc.. These notations are placed beside (main text) **Figure 2 (b)** to accommodate page limit requirements. In the revised version, we will reorganize the figures to arrange them more closely to **Section 2.2**.
>
> #### 3. The source of the result isn’t reported alongside the performance figures presented in the table. Potential inconsistencies in the experimental set-up that may involuntarily skew the comparison.
>
> > The source of data is reported in **Supplementary E**, where we indicate how we obtain the data values and reference the literature where the values are reported. We are aware of the inconsistencies arising from different sources. Therefore, we ensure that we compare against the best baseline values reported. We also acknowledge other time series anomaly detection literature that utilize a similar data populating method [11, 18, 23].
>
> #### 4. Dedicated Related Work section is required in the main paper to better understand how the work fits alongside other reconstruction-based techniques.
>
> > We will include a section for related work in the revised version similar to the following:
> Reconstruction-based techniques have seen a diversity of approaches over the years. The simplest reconstruction technique involves training a separate model sequentially for each channel using UAE [11]. One type of improvement focuses on network architecture, with notable examples like LSTM-VAE [36], MAD-GAN [37], MSCRED [32], OmniAnomaly [9], and TranAD [41]. Additionally, hybrid architectures like DAGMM [35] and MTAD-GAT [38] have been proposed. Another type aims to improve the anomaly score instead of using the original reconstruction error. Designing this anomaly score involves a considerable amount of art, given the high diversity of methods for calculating it across studies. For instance, USAD uses two weighted reconstruction errors [39], OmniAnomaly employs the "reconstruction probability" as an alternative anomaly score [9], MTAD-GAT combines forecasting error and reconstruction probability [38], and TranAD uses an integrated reconstruction error and discriminator loss as the anomaly score [41].
> In the context of network architecture, our method utilizes a straightforward performer-based encoder-decoder structure without incorporating any specialized components. Based on our insight into the point-sequential detection tradeoff, our approach stands out due to the introduction of nominality and induced anomaly scores. This integration enables us to seamlessly combine point-based and sequence-based reconstruction errors, resulting in competitive performance.

---

> > ### Comment · Reviewer_UPqe · 2023-08-18
> > **Acknowledgement**
> >
> > Thank you for the detailed replies to all reviews.
> >
> > I appreciate the additional review of related work presented in your reply. As acknowledged by the authors, there are several refinements and improvements that can be applied to elevate the quality of the paper. As a result, I do not feel inclined to argue for its acceptance, but will raise my score to a borderline accept.

---

> > > ### Comment · Reviewer_m3iW · 2023-08-21
> > > **Acknowledgement**
> > >
> > > I read through the other reviews, author rebuttals, and responses to those. I would still prefer to see experiments on datasets with lower than 4% anomalies. However, I think that, overall, the authors' responses to the reviews is convincing and there is enough in the importance of the problem that they address and the quality of the paper, with the authors' proposed modifications, to warrant publication in NeurIPS.

---

### Official Review · Reviewer_gJSP · 2023-07-05

**Soundness:** 2 fair
**Presentation:** 3 good
**Contribution:** 2 fair
**Rating:** 5
**Confidence:** 4

**Summary:**

This paper proposes an unsupervised time series anomaly detection algorithm called NPSR (Nominality score conditioned TSAD by Point/Sequential Reconstruction), which combines both point-based and sequence-based reconstruction models. Specifically, it proposes a nominality score which is the ratio of a contextual deviation (or in-distribution deviation) to the total deviation (which is assumed to be the sum of the in-distribution deviation and the out-of-distribution deviation). The contextual deviation and the in-distribution deviation is computed by using the sequence-based and point-based reconstruction models. Based on the nominality score and anomaly score (computed using point-based reconstruction model) computed in the neighborhood, the induced anomaly score is further proposed by considering the temporal relationship. Some theoretical results of the proposed algorithm are provided. Experiments on several benchmark time series anomaly detection datasets are performed to demonstrate the performance of the proposed algorithm in comparison with several state-of-the-art algorithms.

**Strengths:**

1.	This paper studies an important and interesting problem, i.e., how to detect anomalies in time series data without label data.
2.	The proposed algorithm combines both point-based and sequence-based reconstruction models, which achieves quite good performance on several time series anomaly detection benchmark datasets.
3.	Some theoretical results are provided for the proposed algorithm.
4.	The paper is generally well written and the presentation is clear.

**Weaknesses:**

1.	Overall, this paper proposes a heuristic-based unsupervised time series anomaly detection algorithm. The overall pipeline in Algorithm looks Ok to me, but from experiments it seems the most important part is the point-based reconstruction. In other words, the rest of the proposed algorithm may be simplified. The ablation studies in Table 3 also partially confirms it.
2.	Flawed experiments. The F1 scores reported in Table 2 for other algorithms, e.g., Anomaly Transformer, are not consistent with results reported in the literature, such as [33]. More justifications are required.
3.	The theoretical results look sound, but may not be useful in practice. In other words, can you justify the value of these results in real-world time series anomaly detection applications?

**Questions:**

1.	In Table 2, the performance of several algorithms, e.g., Anomaly Transformer, is significantly lower than those reported in the literature. More justification is required. In Figure 4(b), the anomalies on the right of t=14800 are not detected by $M_{pt}+M_{seq}$, but detected by $M_{seq}$, resulting in false negative. Can you provide some explanations?
2.	The proposed algorithm is an unsupervised algorithm. Thus, how to adapt the algorithm on different datasets remains not clear to me (besides the threshold used in adjusting the F1).

---

> ### Author Rebuttal · Authors · 2023-08-09
>
> We express our sincere appreciation to Reviewer *gJSP* for their thorough review and valuable insights. The points raised will be duly addressed in the revised version.
>
> #### 1. From experiments it seems the most important part is the point-based reconstruction. In other words, the rest of the proposed algorithm may be simplified.
>
> > Instead of focusing solely on point-based reconstruction, we believe that the most important idea is to achieve a balance between effectively utilizing both point and sequence-based models and carefully selecting the appropriate parameters.
> As an extreme example, we optimize our algorithm on the Mackey-Glass anomaly benchmark (MGAB), which is a univariate time series dataset, and find that sequential-based models can significantly outperform point-based models when using their reconstruction errors as $A(\cdot)$. As in **rebuttal Figure 1**, we see that sequence-based models can correctly identify the anomalies, whereas point-based models did not learn at all. This is reasonable, as point-based models only consider a single time point. In general, the lower the dimensionality of the dataset, the harder it gets for a point-based model to learn statistically meaningful representations.
> Moreover, looking at the high-dimensional datasets reported in (main text) **Table 3**, the induced anomaly score remains important for some datasets: For the MSL dataset, F$1^*$ improves 0.1 (F$1^*=0.467$) compared to using the point-based reconstruction (F$1^*=0.366$). In **rebuttal Figure 3**, F$1^*$ improves 0.047 (F$1^*=0.673$) compared to the point-based anomaly score (F$1^*=0.626$).
> Since the amount of improvement depends on the statistical structure of the test data, it is still useful to consider using both sequential-based and point-based reconstruction in general.
> We will include relevant additional discussions in the revision.
>
> #### 2. Flawed experiments. The F1 scores reported in Table 2 for other algorithms, e.g., Anomaly Transformer, are not consistent with results reported in the literature, such as [33]. In Table 2, the performance of several algorithms, e.g., Anomaly Transformer, is significantly lower than those reported in the literature. More justifications are required.
>
> > In (main text) **Table 2**, we present the best F1 scores *without point-adjustment (PA)*, while [33] reports the scores *with PA*. PA involves adjusting predictions based on the true labels. However, as this method has been shown to be a flawed approach that can overestimate performance in previous studies [11, 23, 24, 34], we do not endorse its use.
> Nevertheless, we have included both PA and non-PA data in **Supplementary D**. Our results demonstrate that even simple heuristic methods can appear to achieve state-of-the-art performance when using PA. We see that our method still achieves competitive performance when using PA, but again, the community is moving away from this flawed metric/approach.
>
> #### 3. The theoretical results look sound, but may not be useful in practice. In other words, can you justify the value of these results in real-world time series anomaly detection applications?
>
> > Let's consider (main text) **Figure 3**. If we set $\theta_N=2$ for SWaT (or $\theta_N=10$ for WADI), which ensures that no anomaly points have a nominality score higher than $\theta_N$, and set $d=1$, then according to **Claim 2**, the performance using the induced anomaly score will surpass that of using the original anomaly score. This shows that if we set a proper $\theta_N$, we can always get a better or equal performance using the induced anomaly score. We can also see from **rebuttal Figure 3 (d)** that **Claim 2** always hold when $d=1$ and $\theta_N$ is set such that no anomaly points have a nominality score above $\theta_N$.
> At present, we are using a certain quantile of the nominality score of the training data (e.g., 98.5%) to set $\theta_N$. Estimating the optimal values for $\theta_N$ and $d$ will be an important future work.
>
> #### 4. In Figure 4 (b), the anomalies on the right of t = 14800 are not detected by $M_{pt}+M_{seq}$, but detected by $M_{seq}$, resulting in false negative. Can you provide some explanations?
>
> > The anomaly section between $t=14800$ and $14900$ contains relatively more contextual than point anomalies. This anomaly is described as *Damage 1 MV 001 and raw water pump to drain Elevated Reservoir tank.*
> When solely considering a time point, $M_{pt}$ struggles to recognize the anomaly. However, $M_{seq}$ can effectively detect anomalous time-dependent relationships, leading to a higher anomaly score using $M_{seq}$. Since the reconstruction error of $M_{pt}$ is used as $A(\cdot)$ in (main text) **Figure 4**, we lose the advantage of effectively utilizing that of $M_{seq}$. This results in an induced anomaly score that is not high enough to reach the threshold. An important future direction would be to explore how to select $A(\cdot)$ among multiple models.
>
> #### 5. The proposed algorithm is an unsupervised algorithm. Thus, how to adapt the algorithm on different datasets remains not clear to me (besides the threshold used in adjusting the F1).
> > The difficulty of choosing these parameters is due to the absence of anomalies in the training data and the real-time nature of the algorithm. However, leveraging domain knowledge allows us to make assumptions about anomalies. The selection of parameters, including the choice between soft or hard gate functions, the value for $d$ and the ratio for $\theta_N$, mostly depends on how we presume the anomaly will occur. For instance, $d$ controls the distance that anomaly scores may propagate. This value should be higher if we presume the average anomaly length is long and vice versa.
> Discussions regarding how to choose a soft or hard gate function can be found in our rebuttal replies to reviewer *UPqe* (*Question 1*). We did not discuss it here due to character limit.
> We will include relevant discussions in the revision.

---

> > ### Comment · Reviewer_gJSP · 2023-08-19
> >
> > Thanks for the clarification and extra experiments, particularly the comparison with Anomaly Transformer, which is one of my core concerns for this paper.
> >
> > Also, the theoretical results confirm the soundness of induced anomaly score in comparison of the original anomaly score.
> >
> > In terms of how to apply the proposed algorithms in practical applications, it is also clearly stated in the rebuttal, though I am not fully statisfied.
> >
> > Overall, the proposed algorithm sound good with theoretical support, but the support cannot benefit the practical usage and adoption of this proposed algorithm. I am not saying it is a flaw, but if it could advance in this direction, it will significantly improve the adoption of the proposed algorithm, instead of simply publishing.
> >
> > I do not feel fully convinced for acceptance as Reviewer UPqe, and current rating may be between 4 and 5.

---

### Official Review · Reviewer_EBhj · 2023-07-06

**Soundness:** 3 good
**Presentation:** 4 excellent
**Contribution:** 3 good
**Rating:** 7
**Confidence:** 4

**Summary:**

This paper proposed a novel idea to handle the time-series anomaly detection problem by calculating the "nominality score"(concerns on this name are in the question part and I'll keep using this name in the following review) and the induced anomaly score. And the F1 score can be mathematically proved to improve using the proposed scores. The authors then constructed the point-based and sequence-based reconstruction models to estimate the anomaly score and experiments on several datasets demonstrate the superior performances.

**Strengths:**

1. The proposed method is novel and provides a new angle to address the anomaly detection problem. Without directly considering how "abnormal" a point is, this work start with thinking what is "normal" and induce anomality score based on the nominality score.
2. The authors provided both theoretical and experimental supports for the proposed methods which are reasonable.
3. The writing of the paper is clear and easy to follow.


**Weaknesses:**

1. A major concern is on the expectation of improvement by using $\hat A(\cdot ;g_{\theta_2})$. While the Claim 1 and Claim 2 as well as the proofs gives a guarantee that using this method won't result in a worse F1 score, it says nothing about the gap with/without the method. Intuitively, the expected improvement may be related to $\alpha$ in Eq. (2), $d$, the function, the number of data, and other factors. The actual case could be that even if the $\alpha$ is larger than 1 by a margin (say \alpha=1.5, 2, or 5), the expected number of $N(t)\leq \theta$ for abnormal points are still a small portion of the whole dataset, making the proposed method useless in practical use. The analysis of a bound or expectation is missing.
2. The correctness of the method relies on several hypotheses, typically: 1) distribution of abnormal points $\Delta x_{t,a}^p$ has a larger variance than that of normal points $\Delta x_{t, n}^p$. This is used to guarantee that nominality score of normal points are larger than abnormal points; 2) the threshold is well selected so that the nominality score of all the normal points (with $y_t=0$) passes the threshold, while a significant number of abnormal points cannot pass the threshold. While the hypotheses are reasonable theoretically, I wonder whether they still hold when the scores are estimated and calculated based on the outputs from neural networks.

**Questions:**

1. I wonder whether the word "nominality" is a misuse, although it is presented across the whole paper. The antonym to the word "anomaly" is "normality", and "nominality" is the noun of "nominal" which means "existing in name only; far below the real value or cost", and I think it's not related to the topic at all.
2. How is the backbone of the model (i.e., the performer-based autoencoder) chosen? Will that cause an unfair comparison between the proposed method and the baselines due to the neural model's approximation ability?
3. The supplementary file mentions how a best threshold is chosen, but in 3.4 ablation study the threshold is set to 98.5% of the nominality score from the training data. Why and how will the choice of different threshold influence the performance (i.e., whether the method is sensitive to the choice of hyperparameters?)

**Limitations:**

No apparent limitations and negative societal impact observed by the reviewer.

---

> ### Author Rebuttal · Authors · 2023-08-09
>
> We express our heartfelt gratitude to Reviewer *EBhj* for conducting a comprehensive review and providing valuable insights. We will take into account the raised points and incorporate them appropriately in the revised version.
>
> #### 1. The analysis of a bound or expectation of improvement for the induced anomaly score is missing.
>
> > To compute the bound or expectation for this scenario, certain statistical assumptions need to be made regarding the time series, as demonstrated in the toy example provided in **Section 2.3**. In the context of unsupervised anomaly detection, we can only observe normal data in the training set, which restricts our ability to model the distribution of normal data. Any deviation from this distribution may be labeled as an anomaly. Due to the lack of constraints on anomalies, establishing a universal bound on the potential improvement becomes highly challenging.
> #### 2. While the hypotheses are reasonable theoretically, I wonder whether they still hold when the scores are estimated and calculated based on the outputs from neural networks.
>
> > The hypotheses will be influenced by the underlying statistical structure of the time series and our choice of threshold, but not by the neural network approximations. By using **(1)**, **(8)**, **(11)**,   Δ**x**$^c_t$, Δ**x**$^p_t$, and $\theta_N$, we can calculate the gate function outputs $g_{\theta_N}(N(\cdot))$, thereby inducing any $A(\cdot)$ and computing a superior $\hat{A}(\cdot)$ (provided that $\theta_N$ satisfies the condition in either **Claim 1** or **2**).
> We approximate Δ**x**$^c_t$, Δ**x**$^p_t$, using neural networks, but the accuracy of these approximations will not affect the claims since there always exists some $\theta_N$ that satisfies the conditions (but it will certainly affect the performance).
> The toy example presented in **Section 2.3** serves as a case study to verify the appropriateness of the nominality score defined in **(1)**. We do not assume that real-world datasets have an i.i.d. normal distribution.
>
> #### 3. Misuse of the word “nominality”
> > We use the term **nominal** due to its usage in terms of measurement (*From a philosophical viewpoint, nominal value represents an accepted condition, which is a goal or an approximation, as opposed to the real value, which is always present.*). We did not use the term **normality** due to its related meaning with the normal distribution (*In statistics, **normality tests** are used to determine if a data set is well-modeled by a normal distribution and to compute how likely it is for a random variable underlying the data set to be normally distributed.*) (Source: Wikipedia)
>
> #### 4. How is the backbone of the model (i.e., the performer-based autoencoder) chosen? Will that cause an unfair comparison between the proposed method and the baselines due to the neural model's approximation ability?
>
> > We opt for performer-based models due to their high efficiency compared to the original transformer model [27]. Changing the architecture is also a viable option. Our approach remains feasible as long as we have both a point-based and sequence-based reconstruction model.
> We do not anticipate that our model will lead to unfair comparisons. Similar to this work, most reconstruction-based baselines [11, 32, 36, 39, Zhou et al.] that utilize neural networks propose a combined workflow from designing the model architecture to calculating the anomaly score. As long as we maintain consistency in data preprocessing and metrics, our comparison method is standardized.
>
> - Bin Zhou, Shenghua Liu, Bryan Hooi, Xueqi Cheng, and Jing Ye. 2019. *BeatGAN: anomalous rhythm detection using adversarially generated time series*. In Proceedings of the 28th International Joint Conference on Artificial Intelligence (IJCAI'19). AAAI Press, 4433–4439.
>
> #### 5. The supplementary file mentions how a best threshold is chosen, but in 3.4 ablation study the threshold is set to 98.5% of the nominality score from the training data. Why and how will the choice of different threshold influence the performance (i.e., whether the method is sensitive to the choice of hyperparameters?)
>
> > To recap, there are two thresholds: $\theta_a$ and $\theta_N$. $\theta_a$ is determined automatically by **Supplementary (3)**. $\theta_N$ is a parameter for $g_{\theta_N}(N(\cdot))$,  when using either a soft **(8)** or hard **(11)** gate function. **Rebuttal Figure 3 (a)** shows the nominality scores vs anomaly scores (from point-based reconstruction errors) for the WADI dataset using the parameters shown in **Supplementary Table 1**. Here, we compare the best F1 scores and corresponding false positive rates using soft/hard gate functions at different $\theta_N$. We see that our method is sensitive to both $\theta_N$ and the choice of gate function. If $\theta_N$ is set too high, it will result in an excessive accumulation of anomaly scores on normal time points, leading to an increase in false positives (**rebuttal Figure 3 (c)**). Conversely, if $\theta_N$ is set too low, no anomaly scores will propagate through time points, and the induced anomaly score will approach the original anomaly score (**rebuttal Figure 3 (b)**). The value of 98.5% for $\theta_N$ is an empirical choice that performs well across most datasets.

---

> > ### Comment · Reviewer_EBhj · 2023-08-14
> > **Re: Rebuttal by Authors**
> >
> > I thank the authors for the detailed responses to my questions. Most of my concerns are well addressed. And I'm sorry that I didn't foresee the difficulty to induce a bound or expectation of the improvement by applying your induced anomaly score. The best result I could obtain under your hypothesis (2)~(4) using the hard gate function (11) with $d=1$ and threshold $\theta$ set to cover 100% anomaly points is in a rather complex form. The improvement seems to be relevant to the dimension of data $D$, ratio of std for normal to abnormal points $\alpha$, the size of the dataset, and the proportion of the anomalies. It seems that the increase of $\alpha$, decrease of anomaly proportion, and sadly decrease of the size of dataset (assume still large) will help improve the result. This means the method may be not effective for extreme large dataset containing many anomalies.
> >
> > I'm not sure whether it will be easier under other assumptions, but I admit this is beyond the scope of this paper while I hope to see further analysis in future work. Figure 3 in the provided pdf in the global response gave me the empirical intuition of the possible improvement.
> >
> > I don't have more concerns at current stage. Thus, I'll raise my score to accept.

---

### Official Review · Reviewer_m3iW · 2023-07-09

**Soundness:** 3 good
**Presentation:** 3 good
**Contribution:** 3 good
**Rating:** 6
**Confidence:** 4

**Summary:**

Basically, this paper aims to consider anomalousness of data points both from a point perspective, which is independent of the temporal relationships in the data, and contextual perspective, which reflects temporal relationships in the data. The paper derives an induced anomaly score that considers both.

**Strengths:**

This paper addresses an important problem of considering both point and contextual anomalies. The paper provides good theoretical background for their work and convincing experimental results, showing particular anomalies that their method finds that are not found by methods that look for either point-based or contextual anomalies.

**Weaknesses:**

The datasets on which testing is performed have rather high anomaly rates. The authors should experiment with some datasets with much lower anomaly rates, perhaps by leaving out some anomalies in the datasets that they use.

Post rebuttal comment:
Thanks to the authors for the responses. While the anomaly rates that the authors use are worth testing, it is quite common to use lower anomaly rates, and such lower rates would constitute a better test of the performance of the new algorithm. With regard to removing anomalies from datasets, one can remove individual time series that are identified as anomalous.

**Questions:**

At the start of section 2.2, $\mathbf{x}_^0$ is part of the observed dataset but is also defined to be $\mathbf{x}_t^c + \Delta\mathbf{x}_t^p$. This cannot be correct. Did you mean to define $\delta\mathbf{x}_t^p$?

**Limitations:**

No limitations are described on the work. Societal impact material is not relevant here. However, a description of future work should be provided based on any patterns in the errors that the presented algorithm makes.

---

Post rebuttal addition:
Thanks to the authors for pointing out the discussions on limitations in the supplementary material. However, such material is critical to understanding the nature of the contribution, and so needs to be in the main paper.

---

> ### Author Rebuttal · Authors · 2023-08-09
>
> We extend our sincere appreciation to Reviewer *m3iW* for conducting a thorough review and offering valuable insights. The raised points will be thoughtfully considered and integrated into the revised version.
>
> #### 1. The datasets on which testing is performed have rather high anomaly rates. The authors should experiment with some datasets with much lower anomaly rates, perhaps by leaving out some anomalies in the datasets that they use.
> > In the context of other time series anomaly detection literature, the datasets utilized in our work are widely used, and the anomaly rates are similar. The following contains a list of literature using the datasets:
> - SWaT – [10, 11, 12, 18, 23, 24, 26, 33, 37, 39, 40, 41] Supplementary [20]
> - WADI – [10, 11, 12, 23, 37, 39, 40, 41] Supplementary [20]
> - PSM – [24, 33]
> - MSL – [9, 11, 18, 23, 24, 25, 33, 38, 39, 40, 41]
> - SMAP – [9, 11, 18, 23, 24, 25, 33, 38, 39, 40, 41]
> - SMD – [9, 10, 11, 23, 24, 25, 33, 39, 41] Supplementary [20]
>
> > In considering removal of some anomalies from existing datasets, it is unclear how to appropriately do that – i.e., results may depend strongly on how missing values are imputed, how to remove sequence related anomalies particularly in multivariate cases, etc. We can note as future work, extensions to explore how detection rates extend to very rare anomaly rate scenarios.
> #### 2. At the start of section 2.2, **x**$^0$ is part of the observed dataset but is also defined to be **x**$^c_t$ + Δ**x**$^p_t$. This cannot be correct. Did you mean to define Δ**x**$^p_t$?
>
> > Yes, we are defining Δ**x**$^p_t$ and expressing that by definition (≜), **x**$^0$ equals **x**$^c_t$ + Δ**x**$^p_t$. We will update the equation and include it in the revision.
>
> #### 3. No limitations are described on the work.
>
> > We respectfully point out that we have presented the limitations in **Supplementary G**.
>
> #### 4. A description of future work should be provided based on any patterns in the errors that the presented algorithm makes.
>
> > In the revised version, we will expand the limitations section in the supplementary to include relevant discussions.
> For instance, in (main text) **Figure 4 (b)**, there exists a false negative between $t=14800$ and $t=14900$ when using the induced anomaly score. According to the WADI dataset, this is a 14.26-minute anomaly described as “Damage 1 MV 001 and raw water pump to drain Elevated Reservoir tank.” When examining a single time point, $M_{pt}$ struggles to recognize the anomaly. However, $M_{seq}$ can detect anomalous time-dependent relationships between the time points. This observation indicates that the section contains relatively more contextual anomalies than point anomalies, resulting in a higher anomaly score using $M_{seq}$. Since we use the reconstruction error of $M_{pt}$ as $A(\cdot)$, we lose the advantage of effectively utilizing the reconstruction error of $M_{seq}$, leading to an induced anomaly score that does not reach the threshold.
> An important future direction would be to explore how to select the model used for calculating $A(\cdot)$.

---

### Author Rebuttal · Authors · 2023-08-09

We sincerely express our gratitude to all the reviewers for their thorough reviews and valuable insights.

A significant hurdle in time series anomaly detection involves effectively modeling time-dependent relationships and detect both contextual and point anomalies accurately. To tackle this issue, we propose an unsupervised framework – *NPSR* – that utilizes both point and sequence-based reconstruction models. We introduce a *nominality score* derived from the ratio of a combined value of the reconstruction errors, and a subsequent derivation of an *induced anomaly score*. We provide theoretical evidence supporting the superiority of the induced anomaly score under specific conditions. Extensive experiments on various public datasets demonstrate that NPSR outperforms most state-of-the-art baselines.

Overall, the reviews provide constructive feedback and praise the paper's novelty and relevance. They also raise concerns regarding experimental comparisons, clarity of presentation, and the need for addressing certain theoretical aspects of the proposed method.

We diligently addressed all the concerns raised by providing ample evidence and the requested results. The raised points will be thoughtfully considered and integrated into the revised version.

Here is the summary of the revisions:
1. **Parameter Selection (Reviewers UPqe, gJSP, EBhj)**: We have thoroughly discussed heuristics for selecting the gate function (**rebuttal Figure 2**), $d$, and $\theta_N$. Additionally, we have provided two additional experimental results (**rebuttal Figure 1** and **3**) to support our viewpoints.
2. **Theoretical Justifications (Reviewers gJSP, EBhj, m3iW)**: We have presented an example where, according to **Claim 2**, the induced anomaly score demonstrates provable superior performance compared to the anomaly score (**rebuttal Figure 3**). Furthermore, we have clarified the conditions under which these hypotheses hold.
3. **Algorithm Justifications (Reviewers gJSP, EBhj)**: To underscore the importance of utilizing both point and sequence-based models, we have included an additional example (**rebuttal Figure 1**). Moreover, we have discussed the fairness of our algorithm in comparison to other studies.
4. **Missing Sections/Figures (Reviewers UPqe, m3iW)**: In response to the feedback, we will add the Related Works section to the main text, and we will extend the Limitations section in the supplementary material to include future works that can mitigate errors made by the presented algorithm.
5. **Flawed Experiments (Reviewer gJSP)**: We have identified a bug with calculating $\theta_a$ for Anomaly Transformer. The bug has been fixed, and the results have been updated. The results for other algorithms remain consistent. Please refer to the **updated Table 2** in the global response. We wish to emphasize that we ***do not use*** the *point-adjustment* method (a method for adjusting predictions according to true labels), which is used in some other literature.
6. **Experiment Results Explanation (Reviewer gJSP)**: In the revision, we have extended our discussion about the false negatives appearing in (main text) **Figure 4 (b)** to provide a more comprehensive explanation.

Additionally, we have noticed that some important information is present in the supplementary material but not in the main text. In response to this, we will rearrange certain aspects in the revision to ensure all pertinent information is appropriately included in the main text.

We are enthusiastic about addressing any further questions or inquiries from the reviewers to ensure the continued improvement and refinement of our work.

---

### Decision · Program_Chairs · 2023-09-21

**Decision:**

Accept (poster)

**Comment:**

This paper addresses a problem of unsupervised time series anomaly detection, presenting a method which leverages both point-based and sequence-based reconstruction. A nominality score is introduced as the ratio of in-distribution deviation of the total deviation and the induced anomaly score is presented. The problem itself is timely and interesting. All reviewers feel that the paper has interesting contributions with some theoretical results as well. The authors did a good job in resolving most of reviewers' concerns in their rebuttal,
leading to the overall score raised. It was suggested to include experiments on datasets with lower than 4% anomalies. I believe that the authors can handle this in their final version.